# Your Pre-trained LLM is Secretly an Unsupervised Confidence Calibrator

**Beier Luo**[1], **Shuoyuan Wang**[1], **Sharon Li**[2], **Hongxin Wei**[1]*
[1]Department of Statistics and Data Science, Southern University of Science and Technology
[2]Department of Computer Sciences, University of Wisconsin-Madison

## Abstract

Post-training of large language models is essential for adapting pre-trained language models (PLMs) to align with human preferences and downstream tasks. While PLMs typically exhibit well-calibrated confidence, post-trained language models (PoLMs) often suffer from over-confidence, assigning high confidence to both correct and incorrect outputs, which can undermine reliability in critical applications. A major obstacle in calibrating PoLMs is the scarcity of labeled data for individual downstream tasks. To address this, we propose Disagreement-Aware Confidence Alignment (DACA), a novel unsupervised method to optimize the parameters (e.g., temperature $\tau$) in post-hoc confidence calibration. Our method is motivated by the under-confidence issue caused by prediction disagreement between the PLM and PoLM while aligning their confidence via temperature scaling. Theoretically, the PLM's confidence underestimates PoLM's prediction accuracy on disagreement examples, causing a larger $\tau$ and producing under-confident predictions. DACA mitigates this by selectively using only agreement examples for calibration, effectively decoupling the influence of disagreement. In this manner, our method avoids an overly large $\tau$ in temperature scaling caused by disagreement examples, improving calibration performance. Extensive experiments demonstrate the effectiveness of our method, improving the average ECE of open-sourced and API-based LLMs (e.g., GPT-4o) by up to 15.08% on common benchmarks.

## 1 Introduction

Post-training has been a critical procedure to ensure large language models (LLMs) generate helpful, honest, and harmless responses [Weng et al., 2023, Kumar et al., 2025]. While post-trained language models (PoLMs) perform well on various downstream tasks [Achiam et al., 2023, DeepSeek-AI and et al., 2025], their reliability and trustworthiness still remain an open challenge. In principle, a reliable LLM should not only demonstrate high confidence in its correct generations but also exercise caution in uncertain situations [Thirunavukarasu et al., 2023, Dahl et al., 2024]. Previous studies [Achiam et al., 2023, Zhu et al., 2023] show that post-training, especially RLHF [Christiano et al., 2017, Stiennon et al., 2020], compromises the well-calibrated confidence estimation of pre-trained language models (PLMs), resulting in over-confidence issues of PoLMs. This gives rise to the importance of confidence calibration for PoLMs, ensuring the confidence score associated with the generation should reflect its ground truth correctness likelihood.

Compared to expensive training methods, post-hoc calibration methods such as temperature scaling [Guo et al., 2017] are more practical for LLMs due to their high efficiency [Shen et al., 2024, Xie et al., 2024]. However, a primary challenge of post-hoc calibration methods is their dependence on labeled data. In practice, generating a reliable labeled dataset for tasks such as mathematics problem solving and medical diagnosis is particularly challenging and time-consuming due to the high level of

---

*Corresponding author (weihx@sustech.edu.cn)

39th Conference on Neural Information Processing Systems (NeurIPS 2025).

domain expertise required. Such difficulty is further compounded by the fact that temperature scaling cannot perform effectively given limited labeled data [Mozafari et al., 2018, Liang et al., 2020]. In contrast, unlabeled data is ubiquitous in real-world deployment scenarios and easy to collect without requiring human intervention. This creates an underutilized resource: vast amounts of unlabeled data are already available during LLM operation, yet are not leveraged for calibration. Thus, this paper studies an unexplored and practical perspective: *How can we achieve effective confidence calibration for PoLMs using unlabeled data in an unsupervised manner?*

To calibrate PoLMs without relying on labeled data, we introduce Disagreement-Aware Confidence Alignment (**DACA**)—a simple and effective post-hoc method that leverages the well-calibrated confidence scores of PLMs. A natural starting point of our idea is to align the confidence of PoLMs with that of PLMs on an unlabeled validation set, minimizing the divergence between the predictive distributions of the PLM and PoLM over all samples. However, we find that this direct confidence alignment can paradoxically lead to under-confidence in the PoLM—when the two models disagree on a prediction, the PLM's confidence often underestimates the actual correctness of the PoLM's output. Our theoretical analysis reveals that such prediction disagreement can drive the optimization to increase the temperature parameter excessively, further exacerbating the under-confidence issue. Motivated by our theory, DACA mitigates this issue by decoupling the influence of disagreement examples from the confidence alignment process. Specifically, it optimizes the temperature parameter using only agreement examples—those where the PLM and PoLM make identical predictions. This ensures that confidence alignment occurs only when the PLM's scores are a trustworthy proxy for correctness. As a result, DACA yields more conservative and reliable temperature estimates, avoiding the calibration failures of naive alignment (see Figure 2b).

Extensive experiments with both open-sourced and API-based LLMs on common benchmarks demonstrate the effectiveness of the DACA method for confidence calibration. Notably, DACA achieves performance comparable to labeled temperature scaling, even in the absence of labeled data. For example, DACA improves the average Expected Calibration Error (ECE) of the Gemma-3-12B-Instruct model [Team et al., 2025] across 57 subjects of the MMLU dataset [Hendrycks et al., 2021], reducing it from 23.68% to 8.60%. In comparison, TS only reduces the ECE to 9.75%. Importantly, DACA is applicable even in scenarios where post-trained and pre-trained models differ in architecture, making it more efficient for the calibration of large-scale PoLMs. For instance, DACA reduces the ECE of GPT-4o [Hurst et al., 2024] from 21.23% to 6.99% when calibrated using the pre-trained Gemma-3-12B model on the MedMCQA dataset [Pal et al., 2022]. Furthermore, our method can be applied to open-ended question-answering tasks and offers benefits for selective classification. Codes are publicly available at https://github.com/ml-stat-Sustech/Disagreement-Aware-Calibration.

We summarize our contributions as follows.

1. We show that the well-calibrated outputs of PLMs on unlabeled data can be leveraged to calibrate PoLMs. Theoretically, we demonstrate that prediction disagreement can impair calibration performance when directly aligning the confidence of PLMs and PoLMs.

2. Our proposed post-hoc method DACA, formalizes the confidence calibration problem by harnessing the target-specific unlabeled data in the wild. This formulation offers strong practicality and flexibility for real-world applications.

3. We empirically show that DACA enhances the calibration of both open-sourced and API-based PoLMs across various datasets. Moreover, our method applies to open-ended QA tasks and enhances selective classification.

## 2 Preliminaries

### 2.1 Confidence Calibration for LLMs

In this work, we focus on the confidence calibration problem of question answering for the Post-trained Language Model (PoLM), denoted as $f$. Our method primarily targets Multiple-Choice QA (MCQA) and can be extended to open-ended QA. For MCQA with choices $\mathcal{Y} = \{A, B, C, D\}$ and prompt $\boldsymbol{x}$, let $z_f(\boldsymbol{x}) \in \mathbb{R}^{|\mathcal{Y}|}$ be the logits of $f$. The predicted probabilities and confidence are

$$p_f(y = j \mid \boldsymbol{x}) = \frac{\exp\left(z_{f,j}(\boldsymbol{x})\right)}{\sum_{j' \in \mathcal{Y}} \exp\left(z_{f,j'}(\boldsymbol{x})\right)}, \qquad \hat{P}(\boldsymbol{x}) = \max_{j \in \mathcal{Y}} p_f(y = j \mid \boldsymbol{x}), \qquad (1)$$

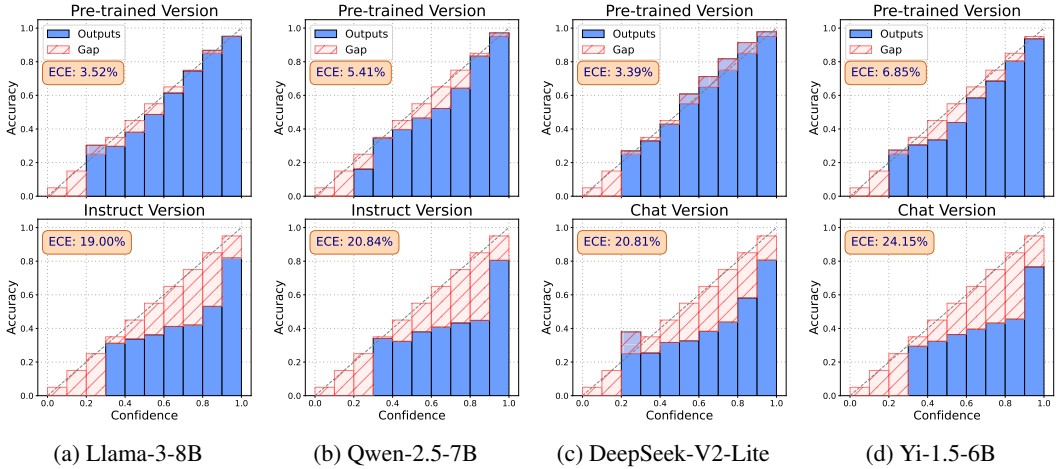

Figure 1: Reliability diagram evaluation for pre-trained vs. post-trained models across four modern LLM architectures on MMLU [Hendrycks et al., 2021]. The post-trained models are trained by multiple post-training techniques, including SFT, RLHF, and DPO. More reliability diagrams of various post-training methods are provided in Appendix A.

with prediction $\hat{Y}(\boldsymbol{x}) = \arg\max_j p_f(y = j \mid \boldsymbol{x})$. We denote the $n$-th prompt in a dataset by $\boldsymbol{x}_n = x_{n,t_n}, ..., x_{n,t_2}, x_{n,t_1}$, which is a sequence of $t_n$ tokens, with its corresponding response denoted as $y_n$. Formally, given a prompt $\boldsymbol{x}$, a perfectly calibrated model satisfies,

$$\Pr(Y = \hat{Y} \mid \hat{P} = \beta) = \beta, \quad \forall \beta \in [0, 1], \tag{2}$$

where $\hat{Y} = \arg\max_y p(y|\boldsymbol{x})$ is the predicted response, and $\hat{P} = \max_y p(y|\boldsymbol{x})$ is the corresponding confidence score [Guo et al., 2017].

To quantify the degree of miscalibration, expected calibration error (ECE) [Naeini et al., 2015] is defined as $\mathbb{E}[|\Pr(Y = \hat{Y}|\hat{P} = \beta) - \beta|]$, which measures the difference between confidence and accuracy. An empirical estimate of ECE is calculated by partitioning $N$ samples into $G$ bins $\{b_1, b_2, \ldots, b_G\}$ according to the confidence predicted by the model. The ECE is then formulated as

$$\text{ECE} = \sum_{g=1}^{G} \frac{|b_g|}{N} \left| \text{acc}(b_g) - \text{conf}(b_g) \right|, \tag{3}$$

where $\text{acc}(b_g)$ and $\text{conf}(b_g)$ denote the average accuracy and confidence within bin $b_g$, respectively. A smaller ECE indicates better calibration performance of the model.

Post-hoc calibration methods aim to calibrate a model after training. Among these approaches, Platt scaling [Platt et al., 1999] based approaches are commonly adopted due to their low complexity and efficiency, including temperature scaling (TS) [Guo et al., 2017] and its extensions [Mozafari et al., 2018, Kull et al., 2019]. In particular, given a miscalibrated model $f$, TS introduces a temperature parameter $\tau$ to soften the model's predicted probability: $p(y = i|\boldsymbol{x}, \tau) = \sigma_i(f(\boldsymbol{x})/\tau)$, where $\sigma(\cdot)$ denotes the softmax function and $\tau > 0$ for all classes. The optimal temperature value for the target dataset by minimizing the negative log-likelihood (NLL) on a labeled calibration dataset $\mathcal{D}^* = \{x_n, y_n\}_{n=1}^{N}$ is given by:

$$\tau^* = \arg\min_{\tau > 0} \left( -\mathbb{E}_{(\boldsymbol{x}, y) \in \mathcal{D}^*} [\log p(y|\boldsymbol{x}, \tau)] \right). \tag{4}$$

Temperature scaling simplifies matrix (vector) scaling [Guo et al., 2017], where a single $\tau$ is applied to all classes, offering great calibration performance while maintaining minimal computational complexity [Guo et al., 2017, Minderer et al., 2021].

## 2.2 The effects of LLM post-training

The success of large language models (LLMs) has led to a standardized training paradigm of pre-training followed by post-training. Post-training refines pre-trained language models (PLMs) for

specific tasks through techniques such as fine-tuning [Ziegler et al., 2019, Wei et al., 2022], alignment [Peng et al., 2023, Su et al., 2023, Bai et al., 2022], knowledge adaptation [Dong et al., 2022, Rubin et al., 2021], and reasoning enhancement [Yao et al., 2023]. While post-training improves task performance, it often comes at the cost of degraded calibration—introducing overconfidence in the model's predictions. In contrast, PLMs typically exhibit more accurate confidence estimates [Achiam et al., 2023, Zhu et al., 2023]. Formally, in multiple-choice tasks, we denote the pre-trained LM as $f : \mathcal{X} \to \mathbb{R}^k$, where $k$ is the number of choices. Through post-training, we learn a post-trained language model (PoLM) $g : \mathcal{X} \to \mathbb{R}^k$. We present the reliability diagram of multiple PoLMs on the MMLU dataset in Figure 1. The diagram illustrates that PoLMs consistently exhibit over-confidence, with confidence scores notably higher than the true likelihood of correctness.

Post-hoc calibration techniques like temperature scaling mitigate overconfidence effectively but rely on labeled validation datasets. Generating a reliable labeled dataset for tasks like mathematical problem-solving and medical diagnosis is challenging and time-consuming due to the required domain expertise. However, under limited labeled data, the calibration performance of post-hoc methods cannot be guaranteed. Leveraging unlabeled data for confidence calibration offers a promising solution for ensuring reliable model behavior in resource-constrained settings. Given the inherently well-calibrated property of PLMs, a natural question arises: *Can we leverage the well-calibrated confidence scores of PLMs on unlabeled data to calibrate over-confident PoLMs?*

## 3 Motivation and Method

To leverage the well-calibrated confidence scores from PLMs, an intuitive approach is to align the confidence levels of PoLMs with those of well-calibrated PLMs on an unlabeled validation set. A naive approach for confidence alignment is to modify the objective in traditional temperature scaling on an unlabeled validation set $\mathcal{D} = \{\boldsymbol{x}_i\}_{i=1}^N$. Instead of minimizing the negative log-likelihood, we minimize the Kullback–Leibler (KL) divergence between the predictive distributions of the pre-trained and post-trained language models on $\mathcal{D}$. Formally, given the post-trained model $g$, the optimal temperature $\tau^*$ on $\mathcal{D}$ is given by

$$\tau^* = \arg \min_{\tau > 0} \mathbb{E}_{\boldsymbol{x} \in \mathcal{D}} \left[ \sum_{i=1}^k p_i(\boldsymbol{x}) \log \frac{p_i(\boldsymbol{x})}{\sigma_i(g(\boldsymbol{x})/\tau)} \right]. \tag{5}$$

Here, $\sigma(\cdot)$ denotes the softmax function, and $p_i(\boldsymbol{x})$ is the $i$-th element of the softmax probability $\sigma(f(\boldsymbol{x}))$ of model $f$. For convenience, we refer to this approach as "naive confidence alignment".

**Naive confidence alignment leads to under-confidence.** In Figure 2a, we show that the naive confidence alignment can lead PoLMs to become significantly under-confident, indicating that their predicted confidence underestimates the actual accuracy. In the following, we investigate why confidence alignment-scaled PoLMs tend to give under-confident predictions. Our analysis suggests that the prediction disagreement introduced by post-training can be a culprit.

Prediction disagreement between two models $f$ and $g$ refers to $\arg \max_i f_i(\boldsymbol{x}) \neq \arg \max_i g_i(\boldsymbol{x})$ on the same input prompt $\boldsymbol{x}$. For convenience, we denote the examples with the existence of prediction disagreement as disagreement examples. It is known that post-training techniques frequently alter the PLM's output distribution, resulting in prediction disagreement. Formally, the unlabeled data can be characterized by the Huber contamination model [Huber, 1992] as follows:

**Definition 3.1** (Unlabeled data distribution). *We define the unlabeled data be the following mixture of distributions*

$$\mathbb{P}_{unlabeled} = (1 - \pi)\mathbb{P}_{agree} + \pi\mathbb{P}_{dis}, \tag{6}$$

*where $\pi \in (0, 1]$ denotes the disagreement ratio, $\mathbb{P}_{agree}$ and $\mathbb{P}_{dis}$ are the marginal distributions of agreement examples and disagreement examples, respectively. In practice, $\pi > 0$, as post-training typically changes some PLMs' predictions.*

With the above definition, we assume the unlabeled dataset $\mathcal{D}$ is i.i.d. sampled from the mixture distribution $\mathbb{P}_{\text{unlabeled}}$. In the following, we analyze the limitations of naive confidence alignment in the presence of prediction disagreement.

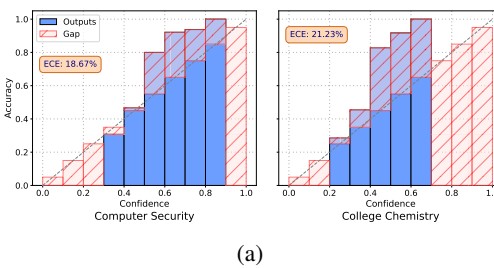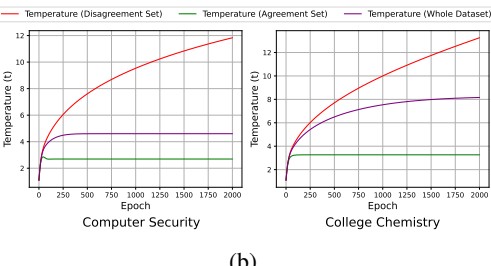

|  (a)  |  (b)  |

Figure 2: Under-confidence issue of naive confidence alignment. (**a**): Reliability diagram for Yi-1.5-9B-Chat on the computer security and college chemistry subjects of MMLU [Hendrycks et al., 2021]. Results of more models are presented in Appendix D. (**b**): Temperature values of Yi-1.5-9B-Chat under different training epochs when trained separately on the disagreement and agreement sets and the whole dataset. The training process is performed on the computer security and the college chemistry subject of MMLU.

**Proposition 3.2.** *Assume $f(\cdot)$ be a perfectly calibrated predictor with $ECE_f = 0$ and $g(\cdot)$ denote a predictor perfectly aligned to the predictor $f$. Let $\tilde{y}$ be the unknown label of sample $\boldsymbol{x}$. The expected calibration error (ECE) of $g$ over the unlabeled distribution $\mathbb{P}_{unlabeled}$:*

$$ECE_g = \pi \cdot \left| \mathbb{E}_{\boldsymbol{x} \sim \mathbb{P}_{unlabeled}} \left[ \mathbf{1}\{\arg\max_i f_i(\boldsymbol{x}) = \tilde{y}\} - \mathbf{1}\{\arg\max_i g_i(\boldsymbol{x}) = \tilde{y}\} \right] \right|.$$

The proposition's proof is presented in Appendix B. The proposition illustrates that the ECE of a PoLM cannot reach zero even in an ideal case where the PoLM is perfectly aligned with a perfectly calibrated PLM in confidence, due to the existence of prediction disagreement. Intuitively, PLM's confidence for disagreement examples reflects its own prediction's accuracy, instead of that of PoLM's prediction. Since post-training typically improves PoLM's accuracy, PLM's confidence level will be lower than the prediction accuracy of PoLM, resulting in the under-confidence issue. In the following, we further analyze how prediction disagreement impacts the parameter $\tau$ of temperature scaling as an example.

**Proposition 3.3.** *Given a sample $\boldsymbol{x}$, let $g(\boldsymbol{x})$ denote the output logits of a post-trained language model, and $\boldsymbol{p}(\boldsymbol{x})$ denote the softmax probability from the pre-trained language model. If $\arg\max_i g_i(\boldsymbol{x}) = c$ and $\sigma_c(f(\boldsymbol{x})) < \frac{1}{k}$, then the optimal temperature is given by:*

$$\tau^* = \arg\min_\tau D_{KL}[\boldsymbol{p}(\boldsymbol{x}) \,\|\, \sigma(g(\boldsymbol{x})/\tau)] = \infty.$$

The proof of this proposition is provided in Appendix B. Proposition 3.3 indicates that the gradient of the KL divergence w.r.t the temperature $\tau$ remains positive on the disagreement set, which increases the value of $\tau$ continuously during optimization. Consequently, the optimization will further exacerbate the under-confidence issue. To provide a straightforward view, Figure 2b shows the temperature dynamics during training exclusively on the disagreement set, revealing a gradual increase to a significantly high value.

**Disagreement-Aware Confidence Alignment.** In our previous analysis, we showed that disagreement examples tend to drive the temperature parameter to excessively high values, leading to an under-confidence issue. To address this problem, our key idea is to decouple the influence of disagreement examples from the confidence alignment process. We propose Disagreement-Aware Confidence Alignment (DACA), which eliminates the gradient of the KL divergence with respect to the temperature on disagreement examples, thereby ensuring that temperature optimization is guided solely by agreement examples. Formally, the new loss function of DACA can be defined as:

$$\mathcal{L}(\tau; \boldsymbol{x}) = \mathbf{1}\{\hat{y} = \hat{y}'\} \cdot \left[ \sum_{i=1}^k p_i(\boldsymbol{x}) \log \frac{p_i(\boldsymbol{x})}{\sigma_i(g(\boldsymbol{x})/\tau)} \right], \tag{7}$$

where $\hat{y} = \arg\max_i f_i(\boldsymbol{x})$ and $\hat{y}' = \arg\max_i g_i(\boldsymbol{x})$ denote the predictions of the pre-trained model $f$ and the post-trained model $g$, respectively.

Minimizing the loss function in Equation (7) mitigates the under-confidence issue effectively. We illustrate with an example in Figure 2b, which demonstrates that optimizing the temperature solely on the agreement set yields a more conservative estimate than optimizing on the whole dataset.

**Extensions to other post-hoc calibration methods.** Notably, our method is general and can be easily incorporated into other existing post-hoc calibration methods such as vector scaling and matrix scaling [Guo et al., 2017]. Formally, for any rescaling function $\phi_{\boldsymbol{\theta}}$ with parameter $\boldsymbol{\theta}$, we can formulate the method as follows. First, we define the $i$-th softmax probability of the post-trained model after rescaling as $q_i(\boldsymbol{x}; \boldsymbol{\theta}) = \sigma(\phi_{\boldsymbol{\theta}} \cdot f(\boldsymbol{x}))_i$. The corresponding probability of the pre-trained model is given by $p_i(\boldsymbol{x})$. Then, the optimization objective can be formulated as:

$$\boldsymbol{\theta}^* = \arg\min_{\tau > 0} \mathbb{E}_{\boldsymbol{x} \in \mathcal{D}} \left[ \mathbf{1}\{\hat{y} = \hat{y}'\} \cdot \sum_{i=1}^{k} p_i(\boldsymbol{x}) \log \frac{p_i(\boldsymbol{x})}{q_i(\boldsymbol{x}; \boldsymbol{\theta})} \right], \tag{8}$$

where $\hat{y} = \arg\max_i p_i(\boldsymbol{x})$ and $\hat{y}' = \arg\max_i q_i(\boldsymbol{x})$ denote the predictions of the pre-trained model and the post-trained model, respectively. We present the calibration performance of our method with vector scaling and matrix scaling in Appendix D.3.

## 4 Experiments

### 4.1 Setup

**Models.** We conduct extensive experiments on diverse LLMs, including both open-source models and those accessible via online APIs. For open-sourced LLMs, we include Llama-3 family [Grattafiori et al., 2024], Gemma-3 family [Team et al., 2025], Qwen-2.5 family [Yang et al., 2024], and Yi-1.5 family [Young et al., 2024]. Unless explicitly stated otherwise, we perform calibration using the pre-trained counterpart of each post-trained LLM. The above models are provided by Hugging Face. To scale up our findings, we also evaluate large-scale LLMs accessed through online APIs, such as GPT-4o [Hurst et al., 2024] and DeepSeek-V3 [Liu et al., 2024a].

**Datasets.** To verify the effectiveness of our proposed methods, we employ three common datasets for evaluations, including: MMLU [Hendrycks et al., 2021], MedMCQA [Pal et al., 2022], and MathQA [Amini et al., 2019]. For MMLU, we learn a specific temperature parameter for each subject using a subject-specific validation set. The datasets are provided by Hugging Face. Due to limited space, detailed information about each dataset is presented in Appendix C.

**Compared methods.** Since our method is the first unlabeled post-hoc approach to calibrate LLMs without training auxiliary models, we exclude many existing calibration methods that rely on labeled data and additional training. To compare with other unlabeled calibration approaches, we select three prompt-based methods as baselines, including **CAPE** [Jiang et al., 2023]: a prompt-based method that calibrates next-token probabilities by permuting option order to mitigate LLM biases, **Elicitation** [Tian et al., 2023]: estimates confidence by prompting the model to generate verbalized probabilities, **Elicitation-Ensemble** [Xiong et al., 2023a]: improves upon this by aggregating outputs from multiple prompts. Specifically, **Vanilla** represents the calibration performance of LLMs without any calibration techniques applied, and Temperature Scaling **(TS)** leverages labeled data from the test task to tune task-specific temperatures and is included as a supervised reference baseline.

**Evaluation metrics.** We evaluate the calibration performance using the following metrics: (1) Expected Calibration Error (**ECE**) [Naeini et al., 2015]: measures the average error between prediction confidence and accuracy across different confidence intervals. For evaluation, we use 10 bins in our evaluation. (2) Maximum Calibration Error (**MCE**) [Naeini et al., 2015]: measures the largest discrepancy between prediction confidence and accuracy across all confidence bins, reflecting the worst-case calibration scenario. (3) Adaptive ECE (**AECE**) [Nixon et al., 2019]: proposes a new binning strategy that uses an adaptive scheme to space the bin intervals, ensuring that each bin contains an equal number of examples. (4) **Brier Score** [Brier, 1950]: directly measures the distance between the model confidence and the binary correctness label of the generation.

**Implementation details.** For multiple-choice datasets, the model estimates the probability that the next token matches one of the options (e.g., A, B, C, or D), reflecting its confidence. Due to the space limitation, more details of implementation are provided in Appendix C.

Table 1: Average calibration performance across 57 MMLU subjects for several contemporary PoLMs. "Vanilla" refers to the uncalibrated model. [†] indicates calibration methods with access to labels. Best results are shown in **bold**, and the second-best results are presented in *italics*. Detailed results for a broader range of LLMs are available in the Appendix D.2.

| Models | Methods | Metrics | | | |
|---|---|---|---|---|---|
| | | ECE %($\downarrow$) | MCE %($\downarrow$) | AECE %($\downarrow$) | Brier Score($\downarrow$) |
| Qwen3-8B | Vanilla | 16.383±0.433 | 38.190±1.547 | 24.990±0.667 | 0.179±0.003 |
| | CAPE | 11.524±0.091 | 31.741±0.152 | 17.614±0.048 | 0.157±0.001 |
| | Elicitation | 16.774±0.214 | 66.884±16.785 | 27.568±2.897 | - |
| | Elicitation-Ensemble | 16.475±0.407 | 44.991±11.249 | 20.515±2.394 | - |
| | Ours | **8.393±0.228** | **23.700±1.374** | **12.601±0.617** | **0.144±0.001** |
| | TS[†] | *8.655±0.220* | *28.108±1.730* | *14.547±0.666* | *0.146±0.001* |
| Gemma-3-12B-Instruct | Vanilla | 23.679±0.525 | 48.506±1.584 | 35.886±1.257 | 0.235±0.005 |
| | CAPE | 13.906±0.209 | 32.830±0.700 | 19.278±0.377 | 0.168±0.001 |
| | Elicitation | 25.464±0.877 | 76.000±15.487 | 41.485±3.731 | - |
| | Elicitation-Ensemble | 25.417±0.244 | 42.017±10.256 | 32.221±1.987 | - |
| | Ours | **8.596±0.380** | **27.022±3.335** | **13.551±0.804** | **0.154±0.002** |
| | TS[†] | *9.746±0.364* | *29.804±2.750* | *15.604±0.859* | *0.159±0.003* |
| Yi-1.5-34B-Chat | Vanilla | 16.200±0.554 | 33.819±1.452 | 20.353±0.664 | 0.199±0.005 |
| | CAPE | 10.251±0.289 | 22.759±0.665 | 13.121±0.012 | 0.179±0.001 |
| | Elicitation | 27.152±6.513 | 83.000±8.000 | 49.211±9.379 | - |
| | Elicitation-Ensemble | 23.954±7.487 | 61.487±11.487 | 39.259±3.049 | - |
| | Ours | *9.465±0.174* | **19.898±1.082** | **11.700±0.411** | *0.174±0.004* |
| | TS[†] | **8.592±0.170** | *28.599±1.377* | *12.553±0.378* | **0.173±0.004** |
| Llama-3-70B-Instruct | Vanilla | 12.870±0.483 | 36.873±1.415 | 23.837±0.760 | 0.143±0.003 |
| | CAPE | 9.346±0.122 | 30.903±1.498 | 17.681±0.172 | 0.125±0.001 |
| | Elicitation | 11.227±0.113 | 60.000±14.142 | 21.237±1.036 | - |
| | Elicitation-Ensemble | 16.632±0.068 | 70.066±28.774 | 21.790±6.976 | - |
| | Ours | **7.844±0.418** | **24.275±1.285** | **13.158±0.488** | **0.120±0.001** |
| | TS[†] | *8.360±0.283* | *27.366±1.778* | *14.928±0.686* | *0.126±0.002* |

## 4.2 Main results

**DACA significantly improves the calibration performance of PoLMs.** Table 1 presents the average calibration performance of the baselines and our method across 57 subjects of the MMLU datasets, with four contemporary LLMs. The validation set is the validation split of each subject in MMLU on Huggingface, where the size of the validation set is limited. A salient observation is that our method effectively mitigates the mis-calibration in various models across all metrics and is even comparable to the labeled TS with limited validation data. For instance, our method improves the ECE of Llama-3-70B-Instruct from $12.870\%$ to $7.844\%$. Similarly, it improves the ECE of the latter released Qwen3-8B from $16.383\%$ to $8.566\%$. It is worth noting that the verbalization-based method, such as Elicitation and Elicitation-Ensemble, performs significantly worse than the next-token logits-based method, which is consistent with the results reported in previous work [Shen et al., 2024]. We further evaluate our method on additional datasets, including MedMCQA and MathQA, as shown in Appendix D.2. Our method can also be extended to vector and matrix scaling, with results shown in Appendix D.3, demonstrating improved calibration across these post-hoc methods.

**DACA is effective across models of different sizes.** We also verify the calibration performance of the baselines and our methods from models of different sizes. In Figure 3, our results indicate that our approach is effective with different-sized LLMs and achieves impressive performance across diverse architectures. Notably, the Vanilla ECE decreases monotonically with increasing model scale, a trend that aligns with the conclusions drawn in previous research [Zhu et al., 2023].

**DACA is agnostic to the choice of PLMs.** In practice, many closed-source, large-scale PoLMs (e.g., GPT-4o and DeepSeek-V3) are accessed via APIs. As such, calibrating these API-based models becomes essential. However, these models typically lack accessible pre-trained versions, and their large scale requires significant computational resources. Our method effectively calibrates both API-based and large-scale PoLMs, as well as smaller models. Specifically, we use three small-scale PLMs—Llama-3-8B, Qwen2.5-7B, and Gemma-3-12B—to calibrate GPT-4o and DeepSeek-V3. As shown in Table 2, our method consistently improves the calibration performance of GPT-4o regardless of the PLM choice. For example, DACA reduces the ECE of GPT-4o from $21.231\%$ to $6.993\%$ using

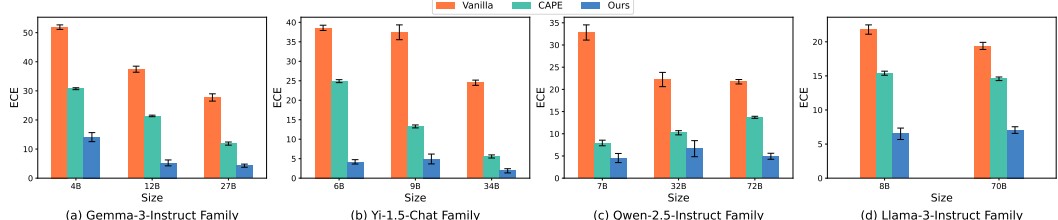

Figure 3: ECE comparison between our methods and baselines on MedMCQA across varying contemporary LLM families and parameter sizes.

Table 2: Calibration performance of DACA for GPT-4o using various pre-trained models on MedM-CQA. "Vanilla" refers to the uncalibrated model. *ECE** represents the original ECE of pre-trained models. Best results are shown in **bold**.

| Methods | Pre-trained Models | Metrics | | | | |
|---|---|---|---|---|---|---|
| | | *ECE*\*% | ECE %($\downarrow$) | MCE %($\downarrow$) | AECE %($\downarrow$) | Brier Score($\downarrow$) |
| Vanilla | - | - | 21.231±0.296 | 35.218±4.260 | 27.619±1.661 | 0.216±0.003 |
| | Llama-3-8B | *9.450±0.777* | 7.984±0.397 | 10.640±0.413 | 6.879±0.737 | 0.150±0.001 |
| Ours | Qwen2.5-7B | *6.990±0.102* | 7.816±0.215 | 10.467±0.42 | 6.751±0.763 | 0.150±0.001 |
| | Gemma-3-12B | *4.424±0.696* | **6.993±0.490** | **10.057±0.115** | **6.115±0.787** | **0.148±0.002** |

Gemma-3-12B. While the calibration performance of PoLMs is similar when scaled with the three PLMs, we find that better-calibrated PLMs yield lower ECEs after alignment. We provide the detailed calibration results for DeepSeek-V3 in Appendix D.4.

**Is our method effective with different post-training strategies?** To demonstrate that our proposed method is agnostic to the post-training strategy, we conduct experiments on a diverse set of Llama-3.1-8B models post-trained with different techniques and report the results in Table 3. We use the models released by Ai2 on Hugging Face[2]. The results show that our method consistently improves calibration performance across all tested post-training strategies. For example, DACA reduces the calibration error of the model post-trained with SFT and DPO from 25.193% to 5.418%. Additional results on post-trained models with different post-training techniques are provided in Appendix D.5.

## 5 Discussion

**Can DACA be applied to open-ended QA tasks?** Previous works estimate confidence scores in open-ended question answering (QA) tasks by reformulating the free-form QA problem into a multiple-choice format [Shen et al., 2024, Kapoor et al., 2024]. Specifically, they pose a binary "Yes" or "No" question to a language model, asking whether its own generated answer is correct or incorrect. This approach, commonly referred to as P(True) in the hallucination detection literature, serves as a well-known baseline. Following prior works, we also adopt the P(True) approach to obtain confidence scores for our experiments. Formally, the confidence score of model $f$ on sample $x$ is defined as $p(\text{Yes}|x, f)$. We then define the prediction disagreement between models $f$ and $g$ in open-ended QA tasks as $\arg\max_i p_i(\boldsymbol{x}, f) \neq \arg\max_i p_i(\boldsymbol{x}, g)$, where $i \in \{1, 2\}$.

Figure 4 illustrates the calibration performance of our method on the TruthfulQA datasets [Lin et al., 2021], evaluated across models of varying sizes from the Qwen2.5 and LLaMA-3 families. Specifically, we use Qwen2.5-32B and LLaMA-3-70B as the pre-trained models to calibrate the corresponding post-trained models within each family. The results demonstrate that our method consistently reduces calibration error across different models. For example, DACA reduces the vanilla ECE from 30.955% to 5.244% on Qwen2.5-32B-Instruct, highlighting its applicability to open-ended QA tasks. Detailed results, including additional metrics, are provided in Appendix D.6.

**DACA can benefit selective classification.** Selective classification [Geifman and El-Yaniv, 2017] leverages model confidence to decide whether to make a prediction or abstain, thereby improving

---

[2]https://huggingface.co/allenai

Table 3: Calibration performance of DACA and baselines on MedMCQA across different post-training techniques applied to Llama-3.1-8B. "Vanilla" refers to the uncalibrated model, while "Oracle TS" represents a lower bound achieved by temperature scaling with access to labeled data from the test task. Best results are shown in **bold**.

| Post-training Techniques | Methods | Metrics | | | |
|---|---|---|---|---|---|
| | | ECE %(↓) | MCE %(↓) | AECE %(↓) | Brier Score(↓) |
| SFT | Vanilla | 14.850±0.857 | 19.893±1.736 | 14.289±0.649 | 0.237±0.004 |
| | CAPE | 7.533±0.334 | 12.323±1.268 | 7.898±0.224 | **0.210±0.001** |
| | Ours | **4.573±0.410** | **10.000±0.000** | **4.812±0.800** | 0.213±0.001 |
| SFT + DPO | Vanilla | 25.120±0.953 | 29.381±1.534 | 22.413±1.387 | 0.282±0.004 |
| | CAPE | 15.576±0.325 | 19.765±1.314 | 14.867±0.835 | 0.233±0.001 |
| | Ours | **5.418±0.354** | **10.000±0.000** | **4.961±0.601** | **0.212±0.001** |
| SFT + DPO + RLVR | Vanilla | 25.193±1.171 | 30.836±1.598 | 22.447±2.532 | 0.282±0.005 |
| | CAPE | 15.729±0.363 | 20.621±1.093 | 14.960±0.925 | 0.234±0.001 |
| | Ours | **5.988±0.430** | **10.000±0.000** | **5.961±0.709** | **0.212±0.001** |

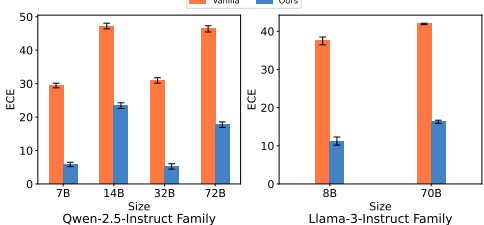

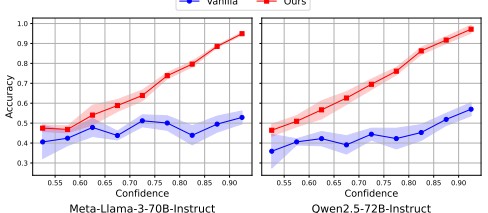

Figure 4: ECE of DACA with different LLMs for the open-ended TruthfulQA benchmark. The lower ECE indicates better calibration performance. Detailed results with more models are provided in Appendix D.

Figure 5: Selective classification accuracy on MedMCQA across different models. Accuracy is reported on subsets of examples with confidence scores above thresholds ranging from 0.55 to 0.95.

reliability by trading off coverage for higher accuracy on accepted examples. This is particularly important when using LLMs for decision-making, where unreliable predictions can lead to significant downstream consequences. Although temperature scaling is accuracy-preserving by design, calibrated confidence scores can nonetheless enhance selective classification by enabling more reliable abstention decisions, thereby improving accuracy on the retained subset.

In Figure 5, we present the accuracy comparison of baselines and our method under varying confidence thresholds ranging from 0.5 to 0.95, where predictions with confidence below the threshold are rejected. A salient observation is that confidence scores calibrated by our method significantly exceed the original accuracy at every confidence threshold, demonstrating improved reliability in selective classification. Notably, the performance gains become increasingly pronounced as the confidence threshold rises. This is attributable to our method's ability to mitigate over-confidence issues, thereby improving the model's accuracy on high-confidence predictions.

# 6 Related work

**Post-training in LLMs.** Post-training in Large Language Models (LLMs) is a critical phase that refines models after their initial pre-training [Tie et al., 2025, Kumar et al., 2025], where they learn general language patterns through next-token prediction on vast datasets. In the post-training phase, LLMs undergo a structured enhancement process that typically follows a sequential order. Initially, fine-tuning is employed to adapt the pre-trained model to specific tasks or domains. This step involves updating the model's parameters using curated datasets, which significantly improves its performance on targeted tasks [Yue et al., 2023, Luo et al., 2023]. To optimize resource efficiency, parameter-efficient fine-tuning (PEFT) techniques, such as Low-Rank Adaptation (LoRA) and Adapters [Hu et al., 2022, Gao et al., 2023, Luong et al., 2024], are often utilized. These methods adjust only a small subset of the model's parameters or introduce a limited number of trainable parameters, achieving comparable performance to full fine-tuning while significantly reducing computational and memory requirements. Following this, reinforcement learning (RL) techniques are applied to

further refine the model's behavior. Methods such as Reinforcement Learning from Human Feedback (RLHF) [Ouyang et al., 2022] and Direct Preference Optimization (DPO) [Rafailov et al., 2023] incorporate dynamic feedback to optimize decision-making and align the model's outputs with user preferences. Together, these strategies transform LLMs into versatile, user-aligned tools for diverse applications. In this work, we address the confidence calibration problem in Post-trained Language Models (PoLMs) by leveraging well-calibrated Pre-trained Language Models (PLMs). Our method aligns the confidence scores of PoLMs with PLMs on samples where both models produce the same prediction.

**Confidence Calibration.** Confidence calibration has been widely studied to ensure that the confidence levels output by models accurately reflect their true performance. To achieve this, the state-of-the-art calibration methods can be categorized into two paradigms: post-hoc methods [Platt et al., 1999, Guo et al., 2017, Mozafari et al., 2018, Kull et al., 2019, Xiong et al., 2023b, Wang et al., 2024] and regularization methods [Müller et al., 2019, Mukhoti et al., 2020, Hebbalaguppe et al., 2022]. For post-hoc calibration, a representative method is temperature scaling [Guo et al., 2017], which learns a single scalar for rescaling the logit. Recently, several studies have investigated calibration in LLMs [Jiang et al., 2023, Xiao et al., 2022, Chen et al., 2022, Liu et al., 2024b], highlighting that post-training often leads to overconfidence. One line of work explores fine-tuning methods to encourage well-calibrated numerical and linguistic verbalized confidence [Lin et al., 2022, Kapoor et al., 2024, Tao et al., 2025], while another focuses on training auxiliary models to predict model confidence [Kadavath et al., 2022, Liu et al., 2024b, Ulmer et al., 2024] or estimate temperature parameters for unseen tasks [Shen et al., 2024]. However, these approaches typically require labeled data and, in some cases, are computationally expensive. Other works [Xie et al., 2024, Tian et al., 2023] examine post-trained LLMs and show that carefully designed prompts can elicit better-calibrated uncertainty estimates. Distinct from prior approaches, our work is the first to leverage unlabeled data for post-hoc confidence calibration, offering both efficiency and flexibility.

# 7 Conclusion

In this paper, we introduce Distance-Aware Confidence Alignment (**DACA**), an unsupervised post-hoc method designed to calibrate overconfident PoLMs. To the best of our knowledge, this is the first approach that uses unlabeled data for the post-hoc calibration of LLMs. The core idea behind DACA is to decouple the influence of prediction disagreement when aligning confidence between PoLMs and well-calibrated PLMs. Specifically, DACA optimizes the temperature parameter using only agreement examples—those in which the PLM and PoLM make identical predictions—ensuring that confidence alignment occurs only when the PLM's scores serve as a trustworthy proxy for correctness. Extensive experiments demonstrate the effectiveness of DACA in calibrating PoLMs across a wide range of models and common datasets. This method can be easily adopted in practical settings, as it can be applied to both open-sourced and API-based LLMs and is computationally efficient.

**Limitations.** Our method involves an additional inference step using pre-trained models, leading to a modest increase in computational cost. Additionally, filtering out disagreement examples may reduce the pool of unlabeled examples available for calibration. However, this trade-off is generally acceptable, given the wide availability of unlabeled data. Future work could explore how to leverage these disagreement examples to further improve calibration.

# Acknowledgment

Beier Luo and Hongxin Wei are supported by the Shenzhen Fundamental Research Program (Grant No. JCYJ20230807091809020). We gratefully acknowledge the support of the Center for Computational Science and Engineering at the Southern University of Science and Technology for our research.

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

Table 4: Post-trained LLM summarization. "Source" refers to the URL indicating the origin or provider of the post-trained LLM.

| Model | Post-training Techniques | Source |
|---|---|---|
| Llama-3.1-Tulu-3-8B-SFT | SFT | https://huggingface.co/allenai/Llama-3.1-Tulu-3-8B-SFT |
| Llama-3.1-Tulu-3-8B-DPO | SFT+DPO | https://huggingface.co/allenai/Llama-3.1-Tulu-3-8B-DPO |
| Llama-3.1-Tulu-3-8B | SFT+DPO+RLVR | https://huggingface.co/allenai/Llama-3.1-Tulu-3-8B |
| Llama-3-8b-Iter-DPO-179k | Iterative-DPO | https://huggingface.co/OpenRLHF/Llama-3-8b-iter-dpo-179k |
| Llama-3-Base-8B-SFT-IPO | SFT+IPO | https://huggingface.co/princeton-nlp/Llama-3-Base-8B-SFT-IPO |
| Llama-3-8B-Self-Instruct-100K | Self-Instruct | https://huggingface.co/Magpie-Align/Llama-3-8B-Self-Instruct-100K |

# Appendix

## A   Over-confidence issue with more post-trained models

In this section, we present evidence that post-training can lead to overconfidence issues with additional PoLMs, as illustrated in Figures 6 and 7. We summarize the PoLMs, along with their post-training technologies and source websites, in Table 4.

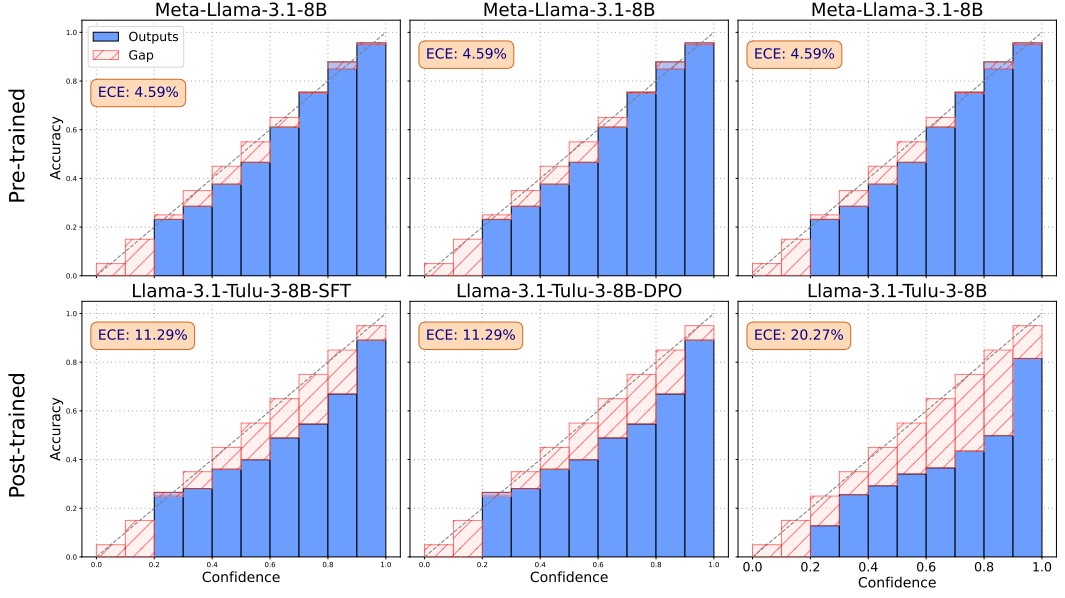

Figure 6: Over-confidence issue of various post-trained Llama-3.1-8B.

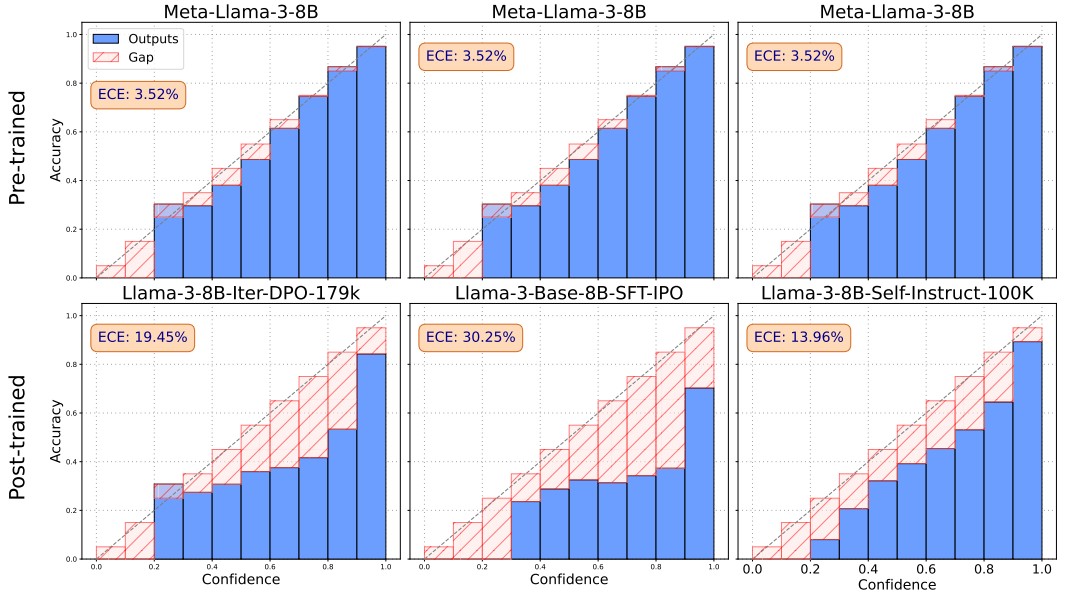

Figure 7: Over-confidence issue of various post-trained Llama-3-8B.

# B  Theoretical Proof

## B.1  Proof of Theorem 3.2

*Proof.* First, we review that the ECE is defined as

$$\text{ECE} = \mathbb{E}\left[\left|\Pr(Y = \hat{Y}|\hat{P} = \beta) - \beta\right|\right].$$

Then given a dataset $\mathcal{D} = \{x_i, \tilde{y}_i\}_{i=1}^N$, the ECE of $f$ is given by

$$\text{ECE}_f = \mathbb{E}_{\boldsymbol{x} \sim \mathbb{P}_{\text{unlabeled}}}\left[p_f(\boldsymbol{x}) - \mathbf{1}\{\arg\max_i f_i(\boldsymbol{x}) = \tilde{y}\}\right],$$

where $p_f(\boldsymbol{x})$ is the confidence score of $f$ on sample $\boldsymbol{x}$. In the same way, the ECE of $g$ is given by

$$\text{ECE}_g = \mathbb{E}_{\boldsymbol{x} \sim \mathbb{P}_{\text{unlabeled}}}\left[p_g(\boldsymbol{x}) - \mathbf{1}\{\arg\max_i g_i(\boldsymbol{x}) = \tilde{y}\}\right],$$

where $p_g(\boldsymbol{x})$ is the confidence score of $g$ on sample $\boldsymbol{x}$. Since the confidence level of $g$ is aligned with $f$, we have that

$$\mathbb{E}_{\boldsymbol{x} \sim \mathbb{P}_{\text{unlabeled}}}\left[p_f(\boldsymbol{x}) - p_g(\boldsymbol{x})\right] = 0.$$

$\square$

## B.2  Proof of Proposition 3.3

*Proof.* The KL divergence between the true distribution $p(x)$ and the model distribution $\sigma(g(x)/\tau)$ is given by:

$$D_{KL}(p(x)||\sigma(g(x)/\tau)) = \sum_{i=1}^{k} p_i(x) \log \frac{p_i(x)}{\sigma(g_i(x)/\tau)}$$

where

$$\sigma(g(x)/\tau)_i = \frac{e^{g_i(x)/\tau}}{\sum_{j=1}^{k} e^{g_j(x)/\tau}}.$$

Our goal is to show that $D_{KL}(p(x)||\sigma(g(x)/\tau))$ is minimized as $\tau \to \infty$.

First, note that the KL divergence can be expressed as:

$$D_{KL}(p(x)||\sigma(g(x)/\tau)) = -H(p(x)) + H(p(x), \sigma(g(x)/\tau)),$$

where

$$H(p(x)) = -\sum_{i=1}^{k} p_i(x) \log p_i(x)$$

is the entropy of $p(x)$, a constant, and

$$H(p(x), \sigma(g(x)/\tau)) = -\sum_{i=1}^{k} p_i(x) \log \sigma(g_i(x)/\tau)$$

is the cross-entropy. Therefore, minimizing $D_{KL}(p(x)||\sigma(g(x)/\tau))$ with respect to $\tau$ is equivalent to minimizing the cross-entropy $H(p(x), \sigma(g(x)/\tau))$.

Next, we analyze the behavior of $\sigma(g(x)/\tau)$ as $\tau$ varies:

- As $\tau \to 0$: Since $c = \arg\max g(x)$, $\sigma(g(x)/\tau)_c \to 1$ and $\sigma(g(x)/\tau)_i \to 0$ for $i \neq c$. If $p_i(x) > 0$ for some $i \neq c$, then $-\log \sigma(g(x)/\tau)_i \to \infty$, implying $H(p(x), \sigma(g(x)/\tau)) \to \infty$.

- As $\tau \to \infty$: $\sigma(g(x)/\tau)_i \to \frac{1}{k}$ for all $i$, since $g_i(x)/\tau \to 0$. Thus,

$$H(p(x), \sigma(g(x)/\tau)) \to -\sum_{i=1}^{k} p_i(x) \log \left(\frac{1}{k}\right) = \log k.$$

Now, for finite $\tau > 0$, since $g_c(x) > g_i(x)$ for $i \neq c$ (assuming a strict maximum for simplicity), we have $\sigma(g(x)/\tau)_c > \frac{1}{k}$, with equality only as $\tau \to \infty$. Given that $p_c(x) < \frac{1}{k}$, the model distribution $\sigma(g(x)/\tau)$ assigns more probability to class $c$ than the uniform distribution for finite $\tau$, while the true distribution $p(x)$ assigns less than uniform to class $c$.

To see why the minimum occurs at $\tau = \infty$, consider that as $\tau$ increases, $\sigma(g(x)/\tau)$ approaches the uniform distribution, which reduces the cross-entropy by making $\sigma(g(x)/\tau)_i$ closer to $\frac{1}{k}$. Since $p_c(x) < \frac{1}{k}$, and typically $p_i(x)$ for $i \neq c$ are such that the uniform distribution provides a better approximation than a distribution concentrated on $c$, the cross-entropy decreases as $\tau$ increases.

More formally, one can consider the derivative of $H(p(x), \sigma(g(x)/\tau))$ with respect to $\tau$, but the limit behaviors suffice to establish that $H(p(x), \sigma(g(x)/\tau))$ is minimized as $\tau \to \infty$. Specifically, since $H(p(x), \sigma(g(x)/\tau)) \to \infty$ as $\tau \to 0$ and $H(p(x), \sigma(g(x)/\tau)) \to \log k$ as $\tau \to \infty$, and assuming $H(p(x), \sigma(g(x)/\tau))$ is continuous and decreasing in $\tau$, the infimum is achieved as $\tau \to \infty$.

Therefore, the temperature parameter that minimizes the KL divergence is:

$$\tau^* = \infty.$$

$\square$

## C  Implementation details

**Experiment details.** We run our experiments on NVIDIA GeForce RTX 4090 and NVIDIA L40 GPU, and implement all methods by *PyTorch* and *vLLM*.

**Optimizer details.** For both TS and DACA, we use the Adam optimizer with a batch size of 256, a learning rate of 0.05, and train for 400 epochs.

**Datasets details.** For the main experiments, we apply confidence calibration to each of the 57 subjects from MMLU and report the average of the calibration metrics. Specifically, we use the validation split of each subject as the validation set. For the MMLU datasets, we conduct five experiments with five different prompts to calculate the mean and standard deviation of the results, as the validation and test splits are predetermined. We provide the choices of the prompt in Table 5. For other datasets, we use the first prompt and report the mean and standard deviation over five random splits of the validation and test sets, with a test-to-validation ratio of 7:3.

Table 5: Variants of multiple-choice question instructions.

| ID | Prompts |
|---|---|
| 1 | The following are multiple-choice questions. Give ONLY the correct option, no other words or explanation: |
| | [Question] A: [Option 1] B: [Option 2] C: [Option 3] D: [Option 4] Answer: [Mask] |
| 2 | Answer the following multiple choice questions by selecting ONLY the correct option: |
| | [Question] A: [Option 1] B: [Option 2] C: [Option 3] D: [Option 4] Answer: [Mask] |
| 3 | For each of the following multiple choice questions, provide just the correct letter: |
| | [Question] A: [Option 1] B: [Option 2] C: [Option 3] D: [Option 4] Answer: [Mask] |
| 4 | Select the correct answer for each of the following questions: |
| | [Question] A: [Option 1] B: [Option 2] C: [Option 3] D: [Option 4] Answer: [Mask] |
| 5 | Choose the right option for each multiple-choice question below. Respond with the letter only: |
| | [Question] A: [Option 1] B: [Option 2] C: [Option 3] D: [Option 4] Answer: [Mask] |

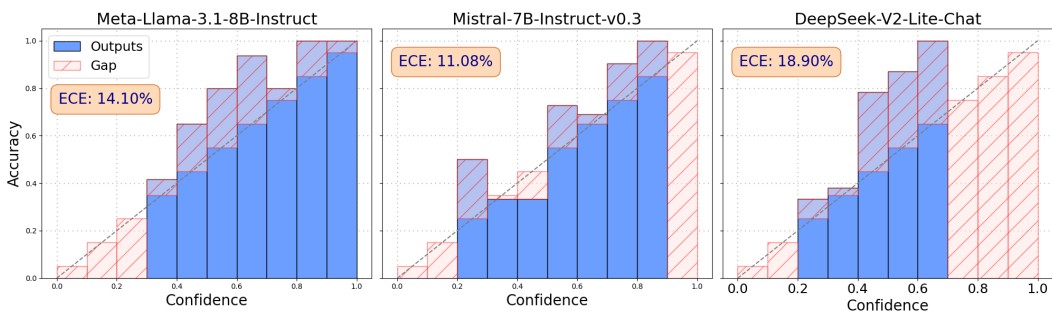

Figure 8: Under-confidence problems of naive confidence alignment with more LLMs.

# D   Detailed results

## D.1   More under-confidence results of naive confidence alignment

We present the reliability diagram of more models scaled with naive confidence alignment on the MMLU dataset in Figure 8.

## D.2   Extended results across diverse models and datasets

**The performance of our method on more datasets and models.**   We present the average calibration performance with more models across 57 subjects of MMLU in Table 6. In addition, we compare the calibration results of our method and baseline approaches on MathQA and MedMCQA in Table 7 and Table 8, respectively. The results show that our method significantly reduces the miscalibration of PoLMs and achieves performance comparable to TS, which has access to labels. For instance, our method reduces the ECE of DeepSeek-V2-Lite-Chat on MedMCQA from 26.553% to 1.715%, while TS reduces to 1.800%.

## D.3   Extension to vector scaling and matrix scaling

We present the results of applying the DACA extension with vector scaling (VS) and matrix scaling (MS) on MedMCQA in Table 9. Across all models, our method consistently reduces the calibration error, regardless of whether VS or MS is used. For example, on Qwen2.5-72B-Instruct, DACA+VS reduces the ECE from 21.720% to 4.133%, which is comparable to the oracle VS result of 4.558%. Similarly, DACA+MS lowers the ECE to 4.407%, closely matching the oracle MS result of 4.201%.

### D.4 Results of additional large-scale PoLMs

We present the results of our method on DeepSeek-V3 with various PLMs in Table 10. Across all PLMs, our method consistently reduces calibration error. A similar trend is observed, where the lower ECE of the pre-trained model leads to a lower ECE in the scaled post-trained model.

### D.5 Results of additional post-training techniques

To evaluate the effectiveness of our method, we perform experiments with more post-trained models, each trained using different post-training techniques. The specific post-training methods applied to each model are listed in Table 4. We present the calibration performance results in Table 11.

### D.6 Detailed results for open-ended tasks

For open-ended tasks, we conduct experiments with Qwen2.5 family and Llama-3 family on the TruthfulQA datasets. For the Qwen2.5 family, we choose Qwen2.5-32B as pre-trained models to calibrate all size post-trained models. And for the Llama-3 family, we choose Llama-3-70B as pre-trained models to calibrate all size post-trained models. We present the detailed results in Table 12 to verify the effectiveness of our method.

Table 6: Average calibration performance across 57 subjects of MMLU on several modern LLMs. "Vanilla" denotes the performance without any calibration applied. [†] represents the calibration method with access to labels. Best results are shown in **bold**, and the second-best results are presented in _italics_.

| Models | Methods | Metrics | | | |
|---|---|---|---|---|---|
| | | ECE %($\downarrow$) | MCE %($\downarrow$) | AECE %($\downarrow$) | Brier Score($\downarrow$) |
| Qwen2.5-7B-Instruct | Vanilla | 21.009 | 39.298 | 30.198 | 0.215 |
| | CAPE | 8.965 | _24.858_ | 14.186 | _0.155_ |
| | Elicitation | 17.962 | 72.867 | 24.998 | - |
| | Elicitation-Ensemble | 26.714 | 43.145 | 23.635 | - |
| | Ours | **7.978** | **23.254** | **10.869** | **0.153** |
| | TS[†] | _8.738_ | 26.747 | _13.720_ | 0.157 |
| Llama-3-8B-Instruct | Vanilla | 17.810 | 36.636 | 22.848 | 0.211 |
| | CAPE | 15.436 | 31.476 | 19.726 | 0.199 |
| | Elicitation | 26.524 | 28.009 | 18.211 | - |
| | Elicitation-Ensemble | 29.548 | 19.794 | 34.334 | - |
| | Ours | _9.485_ | **21.650** | **12.120** | _0.176_ |
| | TS[†] | **8.335** | _25.260_ | _12.246_ | **0.174** |
| Llama-3.1-Tulu-3-8B | Vanilla | 19.977 | 37.794 | 24.551 | 0.219 |
| | CAPE | 11.114 | 24.717 | 15.530 | 0.179 |
| | Elicitation | 27.604 | 38.636 | 27.709 | - |
| | Elicitation-Ensemble | 25.486 | 38.636 | 27.709 | - |
| | Ours | _8.580_ | **19.389** | **11.405** | _0.172_ |
| | TS[†] | **8.475** | _22.336_ | _11.914_ | **0.170** |
| Yi-1.5-6B-Chat | Vanilla | 24.717 | 41.613 | 28.256 | 0.259 |
| | CAPE | 13.183 | _27.348_ | 16.761 | 0.197 |
| | Elicitation | 38.769 | 44.550 | 21.719 | - |
| | Elicitation-Ensemble | 31.504 | 39.339 | 25.478 | - |
| | Ours | _9.208_ | **21.059** | **12.459** | 0.187 |
| | TS[†] | **8.998** | 34.684 | _13.084_ | _0.188_ |
| Yi-1.5-9B-Chat | Vanilla | 22.010 | 40.400 | 28.689 | 0.228 |
| | CAPE | 9.522 | 37.144 | 16.205 | 0.173 |
| | Elicitation | 34.800 | 57.500 | 33.965 | - |
| | Elicitation-Ensemble | 22.405 | 47.619 | 19.640 | - |
| | Ours | _8.814_ | **22.951** | **11.338** | 0.168 |
| | TS[†] | **8.636** | 28.165 | _13.619_ | _0.171_ |
| Mistral-7B-Instruct-v0.3 | Vanilla | 24.860 | 41.401 | 27.878 | 0.259 |
| | CAPE | 13.473 | _26.200_ | 16.899 | 0.198 |
| | Elicitation | 39.840 | 43.549 | 26.308 | - |
| | Elicitation-Ensemble | 34.754 | 50.000 | 29.318 | - |
| | Ours | _9.260_ | **18.385** | **11.554** | _0.186_ |
| | TS[†] | **8.634** | 31.25 | _12.100_ | **0.185** |
| DeepSeek-V2-Lite-Chat | Vanilla | 20.184 | 34.147 | 22.303 | 0.246 |
| | CAPE | 10.219 | **22.348** | 13.745 | **0.197** |
| | Elicitation | 24.483 | 44.466 | 22.999 | - |
| | Elicitation-Ensemble | 27.773 | 34.314 | 23.342 | - |
| | Ours | _9.860_ | _26.590_ | _12.605_ | _0.207_ |
| | TS[†] | **8.661** | 39.242 | **12.538** | _0.207_ |

Table 7: Calibration performance of MathQA on several modern LLMs. "Vanilla" denotes the performance without any calibration applied. [†] represents the calibration method with access to labels. Best results are shown in **bold**.

| Models | Methods | Metrics | | | |
|---|---|---|---|---|---|
| | | ECE %($\downarrow$) | MCE %($\downarrow$) | AECE %($\downarrow$) | Brier Score($\downarrow$) |
| Qwen2.5-7B-Instruct | Vanilla | 35.825±0.826 | 40.369±0.571 | 29.209±0.716 | 0.219±0.001 |
| | Ours | 5.823±0.345 | 22.524±2.420 | **12.589±0.739** | **0.218±0.001** |
| | Oracle TS | **5.511±0.196** | **20.243±1.881** | 12.611±0.808 | 0.219±0.001 |
| Qwen2.5-14B-Instruct | Vanilla | 32.849±0.256 | 36.654±0.535 | 28.628±2.630 | 0.339±0.002 |
| | Ours | **2.795±0.341** | **10.000±0.00** | **4.527±0.166** | 0.220±0.001 |
| | Oracle TS | 5.223±0.236 | 13.881±0.669 | 8.048±0.316 | **0.213±0.001** |
| Qwen2.5-32B-Instruct | Vanilla | 22.239±0.720 | 28.376±1.181 | 21.232±1.190 | 0.257±0.004 |
| | Ours | **3.659±0.390** | 10.001±0.003 | **4.528±0.618** | 0.199±0.001 |
| | Oracle TS | 4.171±0.376 | **10.000±0.000** | 4.655±0.648 | **0.196±0.002** |
| Qwen2.5-72B-Instruct | Vanilla | 30.664±0.346 | 32.900±0.274 | 24.656±1.318 | 0.318±0.003 |
| | Ours | 4.127±0.687 | **10.000±0.000** | **4.278±0.943** | 0.216±0.001 |
| | Oracle TS | **3.472±0.275** | **10.000±0.000** | 4.444±0.223 | **0.212±0.001** |
| Qwen2.5-Math-7B-Instruct | Vanilla | 15.186±0.475 | 26.623±0.939 | 17.473±0.591 | 0.246±0.002 |
| | Ours | 7.491±0.757 | **19.405±6.336** | 10.084±0.911 | 0.225±0.001 |
| | Oracle TS | **3.024±0.596** | 20.324±0.213 | **8.892±5.444** | **0.219±0.001** |

Table 8: Calibration performance of MedMCQA on several modern LLMs. "Vanilla" denotes the performance without any calibration applied. [†] represents the calibration method with access to labels. Best results are shown in **bold**, and the second-best results are presented in _italics_.

| Models | Methods | Metrics | | | |
|---|---|---|---|---|---|
| | | ECE %($\downarrow$) | MCE %($\downarrow$) | AECE %($\downarrow$) | Brier Score($\downarrow$) |
| Qwen2.5-72B-Instruct | Vanilla | 21.814±0.325 | 26.232±0.913 | 26.030±3.165 | 0.237±0.002 |
| | CAPE | 13.488±0.228 | 19.813±0.859 | 15.006±1.677 | 0.187±0.001 |
| | Elicitation | 69.021±0.064 | 70.262±0.449 | 53.556±1.814 | - |
| | Elicitation-Ensemble | 73.151±0.020 | 79.505±0.303 | 35.361±1.772 | - |
| | Ours | **3.938±0.227** | **10.000±0.000** | **4.891±0.683** | **0.173±0.001** |
| | TS[†] | _4.113±0.267_ | **10.000±0.000** | _5.049±0.594_ | _0.174±0.001_ |
| Llama-3-70B-Instruct | Vanilla | 19.814±0.433 | 22.311±1.318 | 20.103±1.358 | 0.217±0.003 |
| | CAPE | 14.272±0.161 | 17.741±1.057 | 18.292±0.356 | 0.188±0.001 |
| | Elicitation | 65.629±0.048 | 71.678±0.377 | 49.057±3.046 | - |
| | Elicitation-Ensemble | 71.147±0.300 | 88.885±7.859 | 41.940±5.357 | - |
| | Ours | **3.464±0.229** | **10.000±0.000** | **4.406±0.537** | **0.163±0.001** |
| | TS[†] | _3.640±0.341_ | 10.000±0.000 | _4.482±0.731_ | 0.163±0.001 |
| DeepSeek-V2-Lite-Chat | Vanilla | 26.553±0.389 | 35.517±0.120 | 23.724±0.460 | 0.311±0.001 |
| | CAPE | 22.414±0.176 | _29.826±0.246_ | 20.5677±0.161 | 0.280±0.001 |
| | Elicitation | 64.193±0.182 | 75.173±0.071 | 44.333±1.003 | - |
| | Elicitation-Ensemble | 63.350±0.435 | 91.219±0.480 | 48.134±1.107 | - |
| | Ours | **1.715±0.357** | 33.521±2.069 | _5.946±1.317_ | **0.229±0.001** |
| | TS[†] | _1.800±0.362_ | **11.506±3.011** | **3.094±0.756** | **0.229±0.001** |

Table 9: Average calibration performance of the DACA extension with vector scaling and matrix scaling on MedMCQA with various models. "Vanilla" denotes performance without any calibration. † denotes methods that are accessible to labels. Best results are shown in **bold**, and the second-best results are presented in *italics*.

| Models | Methods | Metrics | | | |
|--------|---------|---------|---|---|---|
| | | ECE %↓ | MCE %↓ | AECE %↓ | Brier ↓ |
| Llama-3-70B-Instruct | Vanilla | 19.399±0.522 | 22.564±1.356 | 19.574±0.790 | 0.215±0.004 |
| | Ours+VS | 3.838±0.366 | **10.000±0.000** | **5.286±0.831** | *0.164±0.002* |
| | Ours+MS | **3.734±0.413** | *10.067±0.135* | 5.698±1.706 | 0.164±0.003 |
| | VS† | 3.948±0.582 | **10.000±0.000** | *5.685±0.879* | **0.162±0.002** |
| | MS† | *3.823±0.484* | 10.618±1.236 | 6.022±0.877 | **0.162±0.002** |
| Qwen2.5-72B-Instruct | Vanilla | 21.720±0.502 | 28.676±1.605 | 23.413±0.546 | 0.235±0.004 |
| | Ours+VS | **4.133±0.555** | *10.130±0.261* | 5.880±1.376 | 0.175±0.002 |
| | Ours+MS | 4.407±0.665 | **10.086±0.171** | *5.904±1.383* | **0.173±0.002** |
| | VS† | 4.558±0.769 | 10.387±0.775 | 7.038±1.054 | *0.174±0.002* |
| | MS† | *4.201±0.595* | 10.416±0.832 | 6.527±1.196 | *0.174±0.002* |
| Gemma-3-27B-Instruct | Vanilla | 28.914±1.267 | 31.296±1.094 | 24.980±1.767 | 0.303±0.008 |
| | Ours+VS | 4.833±0.792 | **10.000±0.000** | *5.551±0.917* | 0.209±0.003 |
| | Ours+MS | 4.614±0.987 | **10.000±0.000** | 5.904±0.863 | 0.207±0.003 |
| | VS† | **4.409±0.582** | 10.142±0.284 | **5.203±1.046** | *0.202±0.002* |
| | MS† | 5.412±0.580 | *10.089±0.178* | 6.969±0.865 | **0.199±0.003** |

Table 10: Calibration performance comparison of DACA with different pre-trained LLMs on MedMCQA for DeepSeek-V3. "Vanilla" denotes the performance without any calibration applied. Oracle TS serves as the lower bound since it has access to the labeled data for the testing task, and *ECE\** represents the original ECE of the pre-trained model. Best results are shown in **bold**.

| Methods | Pre-trained Models | Metrics | | | | |
|---------|-------------------|---------|---|---|---|---|
| | | *ECE\*%* | ECE %(↓) | MCE %(↓) | AECE %(↓) | Brier Score(↓) |
| Vanilla | - | - | 20.473±0.449 | 29.668±1.588 | 22.518±0.648 | 0.217±0.004 |
| Ours | Llama-3-8B | *9.450±0.777* | 7.127±0.085 | 11.047±0.131 | 6.098±0.085 | 0.161±0.001 |
| | Qwen2.5-7B | *6.990±0.102* | 6.990±0.102 | 10.954±0.082 | 6.071±0.056 | 0.161±0.001 |
| | Gemma-3-12B | *4.424±0.696* | **6.721±0.078** | **10.722±0.074** | **5.855±0.072** | **0.160±0.001** |

Table 11: Calibration performance of MedMCQA of Llama-3-8B post-trained with various techniques. "Vanilla" denotes the performance without any calibration applied. † represents the calibration method with access to labels. Best results are shown in **bold**.

| Post-training Techniques | Methods | Metrics | | | |
|--------------------------|---------|---------|---|---|---|
| | | ECE %(↓) | MCE %(↓) | AECE %(↓) | Brier Score(↓) |
| SFT | Vanilla | 16.225±0.455 | 21.741±0.322 | 15.690±0.472 | 0.244±0.002 |
| | CAPE | 14.286±0.131 | 18.219±0.472 | 14.001±0.913 | 0.227±0.002 |
| | Ours | **6.969±0.255** | **10.000±0.000** | **6.849±0.532** | 0.218±0.001 |
| Iterative-DPO | Vanilla | 23.332±0.261 | 28.756±0.591 | 21.014±1.592 | 0.272±0.003 |
| | CAPE | 19.719±0.167 | 24.740±1.129 | 19.126±0.933 | 0.247±0.002 |
| | Ours | **6.925±0.220** | **10.000±0.000** | **6.701±0.332** | **0.214±0.001** |
| Self-Instruct | Vanilla | 16.981±0.181 | 21.222±0.915 | 15.791±1.314 | 0.242±0.001 |
| | CAPE | 16.379±0.304 | 18.927±0.854 | 16.228±0.852 | 0.231±0.001 |
| | Ours | **7.209±0.306** | **10.486±0.536** | **7.211±0.914** | **0.214±0.001** |

Table 12: Calibration performance on TruthfulQA with several contemporary LLMs. "Vanilla" denotes the performance without any calibration. Best results are shown in **bold**.

| Models | Methods | Metrics | | |
|---|---|---|---|---|
| | | ECE %(↓) | Brier Score(↓) | NLL(↓) |
| Qwen2.5-7B-Instruct | Vanilla | 29.870±1.017 | 0.348±0.007 | 1.155±0.024 |
| | Ours | **6.245±0.974** | **0.253±0.002** | **0.701±0.004** |
| Qwen2.5-14B-Instruct | Vanilla | 47.229±0.847 | 0.479±0.007 | 5.333±0.124 |
| | Ours | **25.423±0.843** | **0.331±0.006** | **0.918±0.016** |
| Qwen2.5-32B-Instruct | Vanilla | 30.955±0.814 | 0.359±0.006 | 1.256±0.025 |
| | Ours | **5.244±0.804** | **0.252±0.002** | **0.698±0.004** |
| Qwen2.5-72B-Instruct | Vanilla | 46.189±1.053 | 0.464±0.009 | 2.889±0.047 |
| | Ours | **17.540±0.916** | **0.277±0.003** | **0.754±0.006** |
| Llama-3-8B-Instruct | Vanilla | 37.615±0.965 | 0.391±0.007 | 1.422±0.040 |
| | Ours | **11.233±1.064** | **0.271±0.003** | **0.739±0.007** |
| Llama-3-70B-Instruct | Vanilla | 42.495±1.160 | 0.430±0.013 | 3.363±0.114 |
| | Ours | **17.001±1.179** | **0.278±0.006** | **0.761±0.014** |

