# OpenReview forum: "Your Pre-trained LLM is Secretly an Unsupervised Confidence Calibrator"
_NeurIPS.cc/2025/Conference — NeurIPS 2025 poster_

### Official Review · Reviewer_uRXE · 2025-06-28

**Clarity:** 3
**Significance:** 2
**Originality:** 3
**Rating:** 4
**Confidence:** 4

**Summary:**

Post-trained large language models (LLMs) are known to suffer from miscalibration after post-training stages such as supervised fine-tuning (SFT) and reinforcement learning with human feedback (RLHF). This paper proposes leveraging a well-calibrated pre-trained LLM to improve the confidence calibration of post-trained LLMs. Rather than optimizing the negative log-likelihood (NLL) loss on a labeled dataset, a standard approach in existing calibration methods, this work proposes minimizing the KL divergence between the predictive distributions of the post-trained and pre-trained LLMs. The authors further argue that this optimization should be restricted to samples on which both models agree in their predictions. Extensive experiments are conducted, demonstrating the effectiveness of the proposed approach across various settings.

**Questions:**

- Instead of performing post-hoc confidence calibration, is it possible to incorporate the KL loss as a constraint during post-training? This could potentially lead to naturally calibrated predictions without relying on post-hoc temperature scaling.


- How well does the proposed method generalize across different tasks? The current approach involves optimizing a temperature parameter for a specific downstream task or domain. Given that different tasks may require different temperature settings, can a **single learned temperature** value effectively calibrate predictions across a diverse set of tasks?

**Ethical Concerns:**

["NO or VERY MINOR ethics concerns only"]

**Final Justification:**

I recommend accepting this paper with rating-4, and the reasons have been provided in my comment to the rebuttal.

**Limitations:**

yes

**Paper Formatting Concerns:**

No major formatting issues are found in this work.

**Quality:**

3

**Strengths And Weaknesses:**

## Strengths
- The paper explores a novel and under-explored setting: unsupervised confidence calibration for LLMs.
- The proposed method is simple yet effective, with well-justified motivation.
- Empirical results are convincing and show consistent improvements across various LLMs.
- The paper is clearly written and easy to follow.

## Weaknesses
- While the problem setting is crucial, the proposed solution may be impractical in real-world scenarios. It requires deploying both the pre-trained and post-trained LLMs and performing inference with both, which could be computationally expensive.
- The method is not universally applicable: the temperature parameter is learned specifically for each downstream task. However, the behavior of post-trained LLMs may vary significantly across tasks, potentially requiring different temperature values.

---

> ### Author Rebuttal · Authors · 2025-07-30
>
> # Response to Reviewer uRXE
>
> Thank you for the constructive feedback. Please find our response below:
>
> ## 1. [W1] Clarification on the inference costs
>
> Thank you for highlighting this concern. We clarify that post-hoc calibration methods, including TS and our DACA, perform optimization (Eq.(7)) during the calibration phase, using a small validation set to tune scaling parameters (e.g., $\tau$). Our method only queries the pretrained model during this phase, not for every test instance. In deployment, DACA incurs no additional computational cost compared to standard post-hoc calibration methods like temperature scaling. Therefore, we can perform inference on test instances using only the post-trained LLM, without needing to deploy or run the pretrained model alongside it. As for the calibration phase, practitioner can first generate the target confidences with (small-scale) PLMs and then use these to tune the scaling parameters for PoLMs, without need to load both models simultaneously. In summary, our solution is practical in real-world scenorios without introducing heavy computational costs.
>
>
> ## 2. [W2, Q2] Generalization of DACA across tasks
>
> Thanks for raising the concern. To evaluate the transferability of the temperature parameter learned by DACA, we conducted experiments by training a temperature value on the MMLU dataset and applying it to the MedMCQA and MathQA datasets. The results in the table below demonstrate that DACA consistently reduces the ECE of several PoLMs on both MedMCQA and MathQA, which validates the practical transferability of the learned temperature $\tau$ (despite it might be not the optimal).
>
>
> | Calibration Dataset | Test Dataset | Model               | ECE (%) Vanilla | ECE (%) DACA | Improvement (%) |
> | :------------------ | :----------- | :------------------ | :-------------- | :----------- | --------------- |
> | MMLU                | MedMCQA      | Llama-3-8B-Instruct | 22.943          | **7.908**    | 15.035          |
> |                     |              | Qwen2.5-7B-Instruct | 32.909          | **19.956**   | 12.953          |
> |                     |              | Yi-1.5-9B-Chat      | 34.250          | **17.775**   | 16.475          |
> |   MMLU                  | MathQA       | Llama-3-8B-Instruct | 42.545          | **20.790**   | 21.755          |
> |                     |              | Qwen2.5-7B-Instruct | 37.266          | **17.197**   | 20.069          |
> |                     |              | Yi-1.5-9B-Chat      | 35.977          | **16.697**   | 19.280          |
>
> Secondly, a more practical scenario is **learning a temperaute value on a few tasks and then generalize to other more tasks**, similar in spirit to the setting of OOD generalization [3]. We conducted experiments by randomly selecting 10 subjects from the MMLU dataset for calibration and testing on the remaining 47 subjects. The table below presents the mean results with standard deviations across 10 random trials. The results demonstrate that DACA consistently improves the calibration performance of three PoLMs across the 47 unseen subjects, indicating DACA's robust generalization to new tasks and robustness to domain shifts.
>
> | Model               | ECE (%) Vanilla | ECE (%) DACA   | Average Improvement (%) |
> | ------------------- | --------------- | -------------- | ----------------------- |
> | Qwen2.5-7B-Instruct | 19.02±0.44      | **10.04±0.28** | 8.98                    |
> | Llama-3-8B-Instruct | 20.26±0.36      | **9.39±0.27**  | 10.87                   |
> | Yi-1.5-9B-Chat      | 21.07±0.52      | **9.57±0.44**  | 11.50                   |
>
> As for learning a single $\tau$ for all 57 subjects in MMLU, we conduct new experiments by learning a single $\tau$ on the validation set of all MMLU subjects and present the average results of test set in the table below. The results show that a single $\tau$ indeed significantly improves the calibration performance of PoLMs on all subjects of MMLU.
>
> | Model               | ECE (%) Vanilla | ECE (%) DACA |Improvement (%) |
> | ------------------- | --------------- | ------------ | ----------------------- |
> | Qwen2.5-7B-Instruct | 18.97           | **9.97**     | 9.00                    |
> | Llama-3-8B-Instruct | 20.22           | **9.40**     | 10.82                   |
> | Yi-1.5-9B-Chat      | 21.18           | **9.85**     | 11.33                   |
>
>
> ## 3. [Q1] Incorporating KL Loss as a post-training constraint
>
> Thank you for this insightful suggestion. We agree that it might be useful to incorporate the KL loss into the post-training process as a regularization. It is worth noting that most popular post-training methods —such as PPO [1] and DPO [2]— indeed employ an explicit or implicit KL-divergence constraint between PoLM and PLM. Their purpose is to improve the training stability by limiting its deviation from the PLM. However, it may introduce a tradeoff: a weaker KL regularization allows the model to better align with preference data, but often at the cost of its calibration performance. Conversely, strong regularization may compromise the performance of preference alignment. As a result, those post-trained models are still poor calibrated (as shown in Table 3), although they are trained with the KL-divergence constraint.
>
> [1] Schulman, John, et al. "Proximal policy optimization algorithms." (2017).
>
> [2] Rafailov, Rafael, et al. "Direct preference optimization: Your language model is secretly a reward model." NeurIPS (2023).

---

> > ### Comment · Reviewer_uRXE · 2025-08-04
> >
> > Thank you to the authors for the rebuttal and the additional experimental results. Overall, I believe this paper is above the acceptance threshold. However, I will maintain my current rating due to some minor concerns regarding the practical value of the work:
> > - The behavior of LLMs can vary significantly across tasks, i.e., can be either over-confident or under-confident. A single learned temperature is unlikely to effectively address both cases.
> > - In practice, applying temperature scaling to calibrate an LLM can negatively impact performance. For instance, reasoning tasks typically require a low temperature to ensure accurate solutions, whereas creative tasks like story writing benefit from a higher temperature to ensure diversity. A single temperature setting cannot generalize well across such diverse scenarios.

---

> > > ### Author Response · Authors · 2025-08-04
> > >
> > > Thank you for reviewing our response and supporting the acceptance. Here, we'd like to provide some clarifications regarding those minor concerns.
> > > > LLMs can be over/under-confident on various tasks
> > >
> > > Yes, we agree that LLMs can exhibit varying behaviors across tasks, where task-specific temperatures are required. This observation motivates our work, which aims to reduce the calibration cost for new tasks by leveraging unlabeled examples. Unlike traditional methods that rely on labeled data to tune task-specific temperatures, our approach uses only unlabeled examples, making it more efficient and scalable for addressing miscalibration in diverse tasks.
> > >
> > > Additionally, we clarify that our work specifically targets the **over-confidence** issue in post-trained LLMs (PoLMs), as highlighted in prior studies [1,2,3], which demonstrate that post-training often exacerbates over-confidence. Our experimental results show that our method consistently improves calibration performance across various tasks, effectively mitigating this issue.
> > >
> > > > Temperature scaling may degrade the LLM performance.
> > >
> > > Thank you for pointing out the potential for confusion. We clarify that our temperature scaling is applied solely for post-hoc confidence calibration, and does not modify the decoding temperature that governs text generation. In particular, this work focuses on logit-based confidences that are widely adopted in LLMs, e.g., GPT-4 [1,2,3,4]. For multiple-choice QA, we derive confidences from the softmax probabilities of the logits corresponding to the selected answer option (e.g., A, B, C, D). For open-ended QA, we adopt the P(True) approach [5], where the model generates an answer and then estimates the probability of its correctness. Since **confidence calibration occurs independently after text generation**, it does not affect the LLM’s performance or behavior during decoding. We believe this approach ensures robust calibration without compromising task-specific performance.
> > >
> > >
> > > [1] Shen, Maohao, et al. "Thermometer: Towards Universal Calibration for Large Language Models." ICML, 2024.
> > >
> > > [2] Xie, Johnathan, et al. "Calibrating Language Models with Adaptive Temperature Scaling." EMNLP, 2024.
> > >
> > > [3] Zhu, Chiwei, et al. "On the Calibration of Large Language Models and Alignment." EMNLP, 2023.
> > >
> > > [4] Achiam, Josh, et al. "Gpt-4 technical report." (2023).
> > >
> > > [5] Kadavath, Saurav, et al. "Language models (mostly) know what they know." (2022).

---

### Official Review · Reviewer_GoW5 · 2025-06-29

**Clarity:** 3
**Significance:** 3
**Originality:** 3
**Rating:** 4
**Confidence:** 5

**Summary:**

This paper introduces Disagreement-Aware Confidence Alignment (DACA), an unsupervised post-hoc calibration method for post-trained language models (PoLMs). The core insight is that while pre-trained language models (PLMs) are often well-calibrated, post-training (e.g., SFT, RLHF) introduces overconfidence. DACA proposes to align the confidence of a PoLM to that of a PLM using only agreement examples—inputs where both models make the same prediction. This strategy avoids calibration degradation due to disagreement examples, which tend to skew temperature scaling toward under-confidence. The method is simple, effective, and does not require labeled data. It significantly improves Expected Calibration Error (ECE) across multiple datasets and LLM families (including closed-source models like GPT-4o), and can generalize to other post-hoc techniques such as vector and matrix scaling.

**Questions:**

1. It can be seen that your results can outperform the TS with access to the ground truth label as shown in table 1, could you give some explanations.

**Ethical Concerns:**

["NO or VERY MINOR ethics concerns only"]

**Final Justification:**

The author has solved most of my concern. However, this work is still limited to limited settings, where training needs the availablility of both pre-trained model and post trained model. But overall, it is a good paper.

**Limitations:**

Yes, the limitations are well included in the paper.

**Quality:**

3

**Strengths And Weaknesses:**

Strengths:
1. The performance is impressive.
2. The table 2 experiment is interesting, it is a surprise to see such an increase by using other PLMs.
3. Empirical results are comprehensive across models, datasets (MMLU, MedMCQA, MathQA), and post-training methods (SFT, DPO, RLVR).
4. Clear experimental baselines and fair comparisons with both supervised and unsupervised methods.


Weaknesses:
1. How you define the confidence distribution of the LLM is important. Should be well defined at the beginning of the analysis. Or people will be lost, how llm expression confidence as traditional models do.
2. It looks like there are some repeat in Eq6 and Eq7.
3. To me, the research setting is strange, it is very limited to the white box and even require access to the pre and post training version. Which is hard to obtain in real practice, let alone the multiple inference of the pre trained model. This issue limited the applicability.
4. The method is limited to multiple option QA.
5. Table 2 is interesting, I wander if you can run some larger open sources PLMs to achieve even better performance.
6. Linguistic Confidence is a trending way to express LLM uncertainty, need to be included and discuss [1,2,3]
7. The method is very limited to the white box and even require access to the pre and post training version. Which is hard to obtain in real practice, let alone the multiple inference of the pre trained model

[1] Revisiting Uncertainty Estimation and Calibration of Large Language Models

[2] Can Large Language Models Faithfully Express Their Intrinsic Uncertainty in Words?

---

> ### Author Rebuttal · Authors · 2025-07-30
>
> # Response to Reviewer GoW5
>
> We appreciate the reviewer for the insightful and detailed comments. Please find our response below:
>
> ## 1. [W1] Definition of LLM confidence distribution
>
> Thank you for this valuable suggestion. In the main part, we focus on the multiple-choice QA task, where an LLM's confidence refers to the softmax probability of the token(s) corresponding to the chosen answer option (e.g., A, B, C, D) [1,2]. In Section 5 (discussion), we extend our method to open-ended QA tasks and employ the P(True) approach [3], where models first generate answers and then assess the probability, P(True), that their responses are correct. Following the suggestion, we will explicitly introduce the definition in Section 2 of the final version.
>
>
> ## 2. [W2] Clarification on the repetition in Eq6 and Eq7
>
> Thank you for the detailed review. We clarify that Eq. (6) defines the loss function for DACA's temperature scaling, while Eq. (7) presents the optimization objective for extending DACA to other post-hoc calibration methods, such as vector scaling (VS) and matrix scaling (MS). Specifically, Eq. (6) is a special case of Eq. (7), where the parameter $\theta$ represents either a scalar temperature $\tau$ for temperature scaling (TS) or vector/matrix parameters for VS and MS. Without the generalized objective in Eq. (7), the extension to various calibration methods might be ambigurous to readers.
>
>
> ## 3. [W3, W7] Clarification on applicability
>
> Thank you for raising this concern. Below, we address each point regarding the applicability of our method with clear explanations.
>
> * **"white-box"**. This work focuses on calibrating logit-based confidence [1,2,3,4], which is accessible in both open-source and many API-based LLMs (e.g., GPT-4o, Grok-4 and Llama-4-Maverick). Therefore, our method is broadly applicable to various LLMs (including APIs) with access to logits.
> * **"require access to the pre and post training version"**. We clarify that the effectiveness of DACA is agnostic to the choice of PLMs (See Tables 2 and 10), so **we do not require access to its pretraining version** for calibrating a PoLM. For example, we can leverage a small-scale, open-sourced PLMs (e.g., Qwen2.5-7B) to calibrate large-scale PoLMs (and APIs) such as GPT-4o and Deepseek-V3. This approach reduces the ECE of GPT-4o on MedMCQA from 21.23% to 7.82%, demonstrating that access to the pretrained version of the PoLM is not required.
> * **"multiple inference of the pretrained model"**. We clarify that post-hoc calibration methods, including TS and our DACA, perform optimization (Eq.(7)) during the calibration phase, using a small validation set to tune scaling parameters (e.g., $\tau$). Our method only queries the pretrained model during this phase, not for every test instance. In deployment, DACA incurs no additional computational cost compared to standard post-hoc calibration methods like temperature scaling.
>
>
> ## 4. [W4] Clarification on the limitation to multiple option QA
>
> In Section 5 (Discussion), we extend the proposed method to open-ended QA tasks. Specically, we employ the P(True) approach [4], where models first generate answers and then assess the probability, P(True), that their responses are correct. Here, "Agreement" occurs when both models give consistent "Yes" or "No" answers. Figure 4 illustrates the effectiveness of DACA in these tasks, notably reducing the Expected Calibration Error (ECE) of Qwen2.5-32B-Instruct on TruthfulQA from 30.955% to 5.244%.
>
> ## 5. [W5] Additional results with larger PLMs
>
> Thank you for this valuable suggestion. We conduct new experiments by employing larger PLMs to calibrate GPT-4o, and present the results on MedMCQA in the table below (including the original results of Table 2 for convenience). The results indicate that leveraging more powerful PLMs can indeed further enhance the calibration of the PoLM.
>
> | Pre-trained Model | ECE (%)  Vanilla | ECE (%) DACA    |
> | :---------------- | :--------------- | :-------------- |
> | Gemma-3-12B       | 21.231±0.296     | 6.993±0.490     |
> | Gemma-3-27B       | 21.231±0.296     | **3.547±0.057** |
> | Llama-3-8B        | 21.231±0.296     | 7.984±0.397     |
> | Llama-3-70B       | 21.231±0.296     | **4.284±0.052** |
> | Qwen2.5-7B        | 21.231±0.296     | 7.816±0.215     |
> | Qwen2.5-72B       | 21.231±0.296     | **4.190±0.106** |
>
> ## 6. [W6] Discussion of Linguistic Confidence
>
> Thank you for introducing the related works. We agree that "Linguistic Confidence" is an interesting and promising approach to expressing LLM uncertainty. In this work, we focus on calibrating logit-based confidence and reveal the benefits of PLMs in confidence calibration. For future work, it will be an interesting direction to design calibration methods for verbalized or linguistic confidence. We will include a discussion of this concept with proper references in Related work of our final version.
>
> ## 7. [Q1] Why DACA can outperform TS with labels?
>
> Thank you for the insightful question. We conjecture that this pheonomenon might be related to the size of the calibration set. Previous studies [5,6] have shown that Temperature Scaling (TS) performs poorly with limited calibration data. Table 1 presents the average performance across 57 MMLU subjects, where TS’s scaling parameter $\tau$ is separately tuned on each subject’s validation set. Many subjects have small validation sets (e.g., $\leq 20$ samples), leading to degraded performance of TS. To validate this, we report the performance of Qwen2.5-7B-Instruct using TS and DACA across subjects with varying calibration set sizes in the table below. The results confirm that DACA outperforms TS in subjects with limited calibration samples, while TS excels when abundant data is available. This highlights DACA’s superior data efficiency, making it particularly valuable in data-scarce scenarios.
>
>
>
> | Subject                   | Size of Calibration Set| Agreement Samples | ECE (%) Vanilla | ECE (%) TS$^\dagger$ | ECE (%) DACA |
> | ------------------------- | ----------------------- | ----------------------- | --------------- | ---------- | ------------ |
> | Machine Learning          | 11      |         4       | 34.428          | 11.859     | **6.943**    |
> | College Mathematics       | 11       |           4    | 36.303          | 17.051     | **9.302**    |
> | High School World History | 26        |      25        | 12.114          | **5.404**  | 5.851        |
> |Professional Law          | 170        |      126       | 38.764          | **6.63**   | 7.926        |
>
> [1] Shen, Maohao, et al. "Thermometer: Towards Universal Calibration for Large Language Models." ICML (2024).
>
> [2] Achiam, Josh, et al. "Gpt-4 technical report." (2023).
>
> [3] Zhu, Chiwei, et al. "On the Calibration of Large Language Models and Alignment." EMNLP (2023).
>
> [4] Kadavath, Saurav, et al. "Language models (mostly) know what they know." (2022).
>
> [5] Mozafari, Azadeh Sadat, et al. "Attended temperature scaling: a practical approach for calibrating deep neural networks." (2018).
>
> [6] Liang, Gongbo, et al. "Improved Trainable Calibration Method for Neural Networks on Medical Imaging Classification." BMVC (2020).

---

### Official Review · Reviewer_APYD · 2025-06-30

**Clarity:** 3
**Significance:** 2
**Originality:** 2
**Rating:** 4
**Confidence:** 3

**Summary:**

This paper introduces an unsupervised approach, Disagreement-Aware Confidence Alignment (DACA), to alleviate the overconfidence problem of post-trained language models. Motivated by the intrinsic weakness of temperature scaling, DACA ignores the disagreement samples when optimizing the temperature parameter, demonstrating competitive performance in extensive experiments and shedding light on efficient post-hoc calibration of large language models.

**Questions:**

1. In the main experiment, 57 different parameters $\tau$ are learned for each subject in MMLU. What about learning a single $\tau$ for all subjects? Is DACA robust under domain shift?
2. Could you specify the values of $\pi$ that were observed in your experiments? Furthermore, have you investigated whether the performance of DACA degrades significantly when this ratio is very high (e.g., on extremely difficult tasks), where the agreement examples may become insufficient or distributionally biased?
3. The definition of `agreement` is ambiguous in real-world LLMs use scenarios. How to improve DACA to help calibrate the confidence of chatbots?
4. Grammar and typos: L.309, You may add an "is" before "computationally efficient".

**Ethical Concerns:**

["NO or VERY MINOR ethics concerns only"]

**Final Justification:**

After careful consideration of their responses and alignment with fellow reviewers, I will maintain my original rating.

**Limitations:**

yes

**Quality:**

3

**Strengths And Weaknesses:**

Strengths:

1. Unsupervised Calibration. As high-quality labeled data is usually expensive to obtain and scale up, DACA explores an unsupervised method for confidence calibration, by innovatively leveraging the well-calibrated confidence of PLMs.
2. Sufficient Theoretical Analysis. This paper reveals that the failure of naive unsupervised calibration stems from disagreement examples between the two models, thereby providing strong theoretical support for the DACA method.
3. Extensive Experiments on Classification. DACA’s effectiveness is grounded by sufficient experiments, and it also performs well on (reformatted) QA and selective classification tasks.

Weaknesses:

1. High dependency on the calibration of PLMs. The ceiling performance of DACA is limited to that of the selected PLM. The concept of "well-calibrated" PLMs is hard to define.
2. Like many other post-hoc schemes, DACA learns a single scalar to adjust the model's output distribution, which does not touch on the underlying confidence mechanism of models, and is likely to cause overfitting. On one task, the learned parameter $\tau$ might be inappropriate to apply to another.
3. As mentioned by the authors, the disagreement samples are directly eliminated in DACA, which may be critical for better calibration.

---

> ### Author Rebuttal · Authors · 2025-07-30
>
> # Response to Reviewer APYD
>
> Thank you for the constructive feedback. Please find our response below:
>
> ## 1. [W1] Dependence on the selected PLMs
>
> Yes, we agree that the calibration level of the chosen PLM may affect DACA's performance. However, prior studies [1,2] demonstrate that most PLMs exhibit significantly better calibration than PoLMs, as validated by Figure 1 of our manuscript. This makes it straightforward to choose an appropriate PLM for implementing DACA. Importantly, the excellent performance of our DACA is not tied to specific PLMs as shown in Table 2. Across various PLMs, including Llama-3-8B, Qwen2.5-7B, and Gemma-3-12B, DACA consistently reduces GPT-4o's ECE from 21.23% to below 8%. This highlights that the choice of PLM does not constrain DACA's practical applicability.
>
> Regarding the term "well-calibrated PLMs" (line 128), we clarify that it refers to the well-calibrated confidence scores of PLMs compared to those of PoLMs, not to restricting PLM selection (as most PLMs perform exceptionally well). In the final version, we will revise the phrase to "align PoLMs' confidence levels with those of PLMs" for greater clarity.
>
>
> ## 2. [W2, Q1] Transferability of DACA
>
>
> Thanks for raising the concern. To evaluate the transferability of the temperature parameter learned by DACA, we conducted experiments by training a temperature value on the MMLU dataset and applying it to the MedMCQA and MathQA datasets. The results in the table below demonstrate that DACA consistently reduces the ECE of several PoLMs on both MedMCQA and MathQA, which validates the practical transferability of the learned temperature $\tau$ (despite it might be not the optimal).
>
>
> | Calibration Dataset | Test Dataset | Model               | ECE (%) Vanilla | ECE (%) DACA | Improvement (%) |
> | :------------------ | :----------- | :------------------ | :-------------- | :----------- | --------------- |
> | MMLU                | MedMCQA      | Llama-3-8B-Instruct | 22.943          | **7.908**    | 15.035          |
> |                     |              | Qwen2.5-7B-Instruct | 32.909          | **19.956**   | 12.953          |
> |                     |              | Yi-1.5-9B-Chat      | 34.250          | **17.775**   | 16.475          |
> |   MMLU                  | MathQA       | Llama-3-8B-Instruct | 42.545          | **20.790**   | 21.755          |
> |                     |              | Qwen2.5-7B-Instruct | 37.266          | **17.197**   | 20.069          |
> |                     |              | Yi-1.5-9B-Chat      | 35.977          | **16.697**   | 19.280          |
>
> Secondly, a more practical scenario is **learning a temperaute value on a few tasks and then generalize to other more tasks**, similar in spirit to the setting of OOD generalization [3]. We conducted experiments by randomly selecting 10 subjects from the MMLU dataset for calibration and testing on the remaining 47 subjects. The table below presents the mean results with standard deviations across 10 random trials. The results demonstrate that DACA consistently improves the calibration performance of three PoLMs across the 47 unseen subjects, indicating DACA's robust generalization to new tasks and robustness to domain shifts.
>
> | Model               | ECE (%) Vanilla | ECE (%) DACA   | Average Improvement (%) |
> | ------------------- | --------------- | -------------- | ----------------------- |
> | Qwen2.5-7B-Instruct | 19.02±0.44      | **10.04±0.28** | 8.98                    |
> | Llama-3-8B-Instruct | 20.26±0.36      | **9.39±0.27**  | 10.87                   |
> | Yi-1.5-9B-Chat      | 21.07±0.52      | **9.57±0.44**  | 11.50                   |
>
> As for learning a single $\tau$ for all 57 subjects in MMLU, we conduct new experiments by learning a single $\tau$ on the validation set of all MMLU subjects and present the average results of test set in the table below. The results show that a single $\tau$ indeed significantly improves the calibration performance of PoLMs on all subjects of MMLU.
>
> | Model               | ECE (%) Vanilla | ECE (%) DACA |Improvement (%) |
> | ------------------- | --------------- | ------------ | ----------------------- |
> | Qwen2.5-7B-Instruct | 18.97           | **9.97**     | 9.00                    |
> | Llama-3-8B-Instruct | 20.22           | **9.40**     | 10.82                   |
> | Yi-1.5-9B-Chat      | 21.18           | **9.85**     | 11.33                   |
>
> ## 3. [W3,Q2] The potential limitation of filtering examples
>
> Yes, the filtered examples could potentially enhance calibration performance, offering a promising avenue for future research. However, this limitation does not affect the contribution of this work, which achieves calibration performance comparable to, or surpassing, supervised temperature scaling. Notably, our method still significantly improve the calibration performance even in cases of high disagreement ratio. The table below presents the results of DACA using Qwen2.5-7B-Instruct (PoLM) and Qwen2.5-7B (PLM) on the three MMLU subjects with the highest disagreement ratios. The results show that our method can outperform supervised TS, even when there are only four agreement samples.
>
>
>
> | Subject             | $\pi$ |Number of Agreement Samples| ECE (%) Vanilla | ECE (%) DACA | ECE (%) TS$^\dagger$ |
> | :------------------ | :----------------- | :--------------| :--------------| :-------------- | :----------- |
> | Machine Learning    | 0.643     | 4         | 34.428      |   **6.943**    |             11.069  |
> | College Mathematics | 0.643      |   4     | 36.303       |   **9.302**    |         21.313 |
> | Abstract Algebra    | 0.452     |    6     | 24.224        |  13.471 | **10.464** |
>
>
> Besides, we report the statistics of disagreement sample ratios $\pi$ across 57 MMLU subjects using three models in the following table.
>
> | Pre-trained Model | Post-trained Model  | Min | Max | Mean ± StD  |
> | :---------------- | :------------------ | :----- |:----- | :----- |
> | Qwen2.5-7B        | Qwen2.5-7B-Instruct | 0 | 0.6364 | 0.180±0.132 |
> | Llama-3-8B        | Llama-3-8B-Instruct | 0 | 0.5 | 0.218±0.127 |
> | Yi-1.5-9B         | Yi-1.5-9B-Chat      | 0.0952 | 0.6923 | 0.337±0.153 |
>
>
> ## 4. [Q3] Definition of `agreement` in open-ended QA
>
> Thank you for the insightful question. In Section 5 (Discussion), we extend the proposed method to open-ended QA tasks. Specically, we employ the P(True) approach [4], where models first generate answers and then assess the probability, P(True), that their responses are correct. Here, "Agreement" occurs when both models give consistent "Yes" or "No" answers. Figure 4 illustrates the effectiveness of DACA in these tasks, notably reducing the Expected Calibration Error (ECE) of Qwen2.5-32B-Instruct on TruthfulQA from 30.955% to 5.244%. Additionally, a potential direction is to define a soft metrics for the `agreement` via measuring the semantic distance between the responses of PLM and PoLM, offering greater flexibility for chatbot applications. This will be included in the final version of our discussion.
>
>
> ## 5. [Q4] Grammar typo
>
> Thank you for pointing this out. We will revise this typo in our final version.
>
> [1] Achiam, Josh, et al. "Gpt-4 technical report." (2023).
>
> [2] Zhu, Chiwei, et al. "On the Calibration of Large Language Models and Alignment." EMNLP (2023).
>
> [3] Liu, Jiashuo, et al. "Towards out-of-distribution generalization: A survey." (2021).
>
> [4] Kadavath, Saurav, et al. "Language models (mostly) know what they know." (2022).

---

> > ### Comment · Reviewer_APYD · 2025-08-05
> >
> > Thank you for your detailed response. I would like to keep my positive score since you have resolved most of my concerns.

---

> > > ### Author Response · Authors · 2025-08-06
> > >
> > > Dear Reviewer APYD,
> > >
> > > Thank you for supporting the acceptance of this paper. We're glad that our responses addressed your concerns. We truly appreciate your valuable time for the reviewing.
> > >
> > > Best regards,
> > >
> > > Authors of submission 11607

---

### Official Review · Reviewer_8hAp · 2025-07-03

**Clarity:** 3
**Significance:** 3
**Originality:** 3
**Rating:** 5
**Confidence:** 4

**Summary:**

This paper propose Disagreement-Aware Confidence Alignment (DACA), a novel unsupervised method to optimize temperature for post-hoc confidence calibration. They theoretically demonstrate that the confidence of pre-trained language models underestimates the prediction accuracy of post-trained language models on disagreement examples and therefore propose DACA to address this issue. The empirical results show that DACA enhances the calibration of both open-sourced and API-based PoLMs across various dataset.

**Questions:**

Please refer to Strengths And Weaknesses.

**Ethical Concerns:**

["NO or VERY MINOR ethics concerns only"]

**Final Justification:**

I maintain my positive score as the authors have addressed most of the concerns.

**Limitations:**

yes

**Quality:**

3

**Strengths And Weaknesses:**

The paper is well-written and well-structured. The related work section clearly outlines the relationship between existing approaches and the proposed method. The authors first theoretically demonstrate that the confidence of pre-trained language models underestimates the prediction accuracy of post-trained language models on disagreement examples. Then they propose a simple yet practical post-hoc calibration method that can be applied to most existing calibration techniques to enhance performance. Therefore, the proposed method is well-motivated. The authors showcase the superior performance of their proposed methods on multiple popular language models and use appropriate and sufficient evaluation metrics to validate the results. I appreciate how the authors highlight their key findings, such as "DACA significantly improves the calibration performance of PoLMs", "DACA is effective across models of different sizes", "DACA is agnostic to the choice of PLMs", and "DACA can benefit selective classification." These concise summaries effectively emphasize the advantages of their proposed approach. Overall, the proposed approach is both theoretically sound and empirically significant. One potential weakness as noted by the authors is that the disagreement examples are filtered out, which reduces the pool of unlabeled examples available for calibration. It would be helpful if the authors could discuss more about how solve the issue.

---

> ### Author Rebuttal · Authors · 2025-07-30
>
> # Response to Reviewer 8hAp
>
> Thanks for your positive and valuable suggestions. Please find our response below:
>
> ## 1. [W1] The potential weakness of filtering examples
>
> Thank you for the thorough review. The filtered examples could potentially enhance calibration performance, offering a promising avenue for future research. However, this limitation does not affect the contribution of this work, which achieves calibration performance comparable to, or surpassing, supervised temperature scaling. Notably, our method still significantly improve the calibration performance even in cases of high disagreement ratio. The table below present the results of DACA using Qwen2.5-7B-Instruct (PoLM) and Qwen2.5-7B (PLM) on the three MMLU subjects with the highest disagreement ratios. The results show that our method can outperform supervised TS, even when there are only four agreement samples. Additionally, a promising direction for utilizing filtered examples is to generate target confidences using specialized agents for confidence alignment, which could be particularly valuable in some extreme cases.
>
> | Subject             | $\pi$ |Number of Agreement Samples| ECE (%) Vanilla | ECE (%) DACA | ECE (%) TS$^\dagger$ |
> | :------------------ | :----------------- | :--------------| :--------------| :-------------- | :----------- |
> | Machine Learning    | 0.643     | 4         | 34.428      |   **6.943**    |             11.069  |
> | College Mathematics | 0.643      |   4     | 36.303       |   **9.302**    |         21.313 |
> | Abstract Algebra    | 0.452     |    6     | 24.224        |  13.471 | **10.464** |

---

> > ### Comment · Reviewer_8hAp · 2025-08-05
> >
> > Thank you for your response. I would like to maintain my positive score.

---

> > > ### Author Response · Authors · 2025-08-06
> > >
> > > Dear Reviewer 8hAp,
> > >
> > > Thank you for supporting the acceptance of this work. We truly appreciate your valuable reviews and suggestions.
> > >
> > > Best regards,
> > >
> > > Authors of submission 11607

---

### Decision · Program_Chairs · 2025-09-17

**Decision:**

Accept (poster)

**Comment:**

The authors propose a novel framework leveraging the pretrained model to calibrate the posttrained model. Reviewers unanimously agree that the paper is above the acceptance bar, though they point out several significant limitations to the method. I recommend acceptance, but the authors will need to incorporate all promised edits to the paper and make sure to thoroughly, carefully, and prominently present the limitations noted by the reviewers in the camera ready.